# Diversity of Algae and Cyanobacteria and Bioindication Characteristics of the Alpine Lake Nesamovyte (Eastern Carpathians, Ukraine) from 100 Years Ago to the Present

Petro M. Tsarenko [1], Olena P. Bilous [2,*], Olha M. Kryvosheia-Zakharova [1], Halyna H. Lilitska [1] and Sophia Barinova [3]

1   M.G. Kholodny Institute of Botany, National Academy of Sciences of Ukraine, Tereschenkivska Str. 2, 01004 Kyiv, Ukraine; ptsar@ukr.net (P.M.T.); olha_krivosheia@ukr.net (O.M.K.-Z.); dunaliella@ukr.net (H.H.L.)
2   Institute of Hydrobiology, National Academy of Sciences of Ukraine, Geroiv Stalingrada 12, 04210 Kyiv, Ukraine
3   Institute of Evolution, University of Haifa, Abba Khoushi Ave, 199, Mount Carmel, Haifa 3498838, Israel; sophia@evo.haifa.ac.il
*   Correspondence: bilous_olena@ukr.net; Tel.: +380-977810018

**Abstract:** The species diversity and changes in the structural dynamics of the algal flora from the alpine lake Nesamovyte has been studied for 100 years. During the period of investigations, 234 species (245 infraspecific taxa) were revealed to cover more than 70% of the modern species composition of the studied lake. The modern biodiversity of algae is characterized by an increase in the number of widespread forms, a change from the baseline "montane" complex in comparison to the beginning of the 20th century. Nevertheless, the Nesamovyte Lake still has a unique algae composition that is typical for high-mountainous European lakes. The presence of a different complex of conventionally arctic species of algae, in particular, diatoms is discussed. Structural changes in the taxonomic composition of the algal flora of the lake as well as in the complex of the leading genera, species and their diversity are revealed. An ecological analysis of the algal species composition of the lake showed vulnerability and degradation to the ecosystem of the lake. On this basis, the issue regarding the question of protection and preservation of the algae significance and uniqueness of the flora of algae in the Nesamovyte Lake are discussed.

**Keywords:** diversity; algae; alpine lake; bioindication; ecological characteristic; ecosystem; Nesamovyte Lake; Eastern Carpathians

## 1. Introduction

The lakes in the mountainous regions of the Eastern Carpathians (as well as in similar areas in the Alpine-Carpathian mountain system) play a leading role in the formation of the algal flora of the region and its conservation of rare species, and are indicators of the local ecosystems. These lakes contain specific regional complexes of algae species that were formed under specific conditions: the origin of the definite waterbodies, their trophic state, water chemistry and the level of anthropogenic load [1–15]. Other comprehensive studies of lakes in the mountain systems of Europe (the Alps, Pyrenees, Balkans, Tatras, Western Carpathians) are considered to be relevant and the results of the studies are used in international projects such as European Mountain lake Ecosystems: Regionalisation, diaGnostics & Socio-economic Evaluation (EMERGY), Alpine Lakes: Paleolimnology and Ecology (AL:PE2), the European Mountain Lake Research (MOLAR) project and many others [6,16–28]. These results have also been implemented in the EU Water Framework Directive 2000/60 [29]. The basis of the paradigm in studying mountain lake ecosystems is to gain knowledge regarding the genesis of waterbodies of this type, stratigraphy and paleolimnology, chemistry and general patterns of their functioning, the presence and

conservation of biotic component and the type of waterbody itself and ecosystems in general, climate impact and its further transformation, as well as socio-economical aspects, etc. However, the basic question still remains regarding the study and analysis of aquatic organisms—their diversity, productivity, relationships, indicators, species specificity and uniqueness. Species diversity of these organisms in general and algae in particular in the Central and Southern European ecosystems of mountain lakes were carefully studied during the end of the XIX–XXth century [30–43]. The species composition of algae found in the mountain lakes in the Carpathian-Tatra region (western part of the Carpathian mountain system) was studied in detail [9,10,35,37,44–60]. However, the eastern part of the region (Eastern Carpathians and especially the Ukrainian Carpathians, which cover an area of 24,000 km$^2$ and have a length of 280 km, from the Polish to the Romanian border) is characterized by incomplete data regarding the algal composition and the lack of information according to individual taxonomic groups and types of waterbodies so far [61]. The situation has improved since the signing in 2014 of the Association Agreement between Ukraine and the European Union and the implementation of the provisions and standards of the Water Framework Directive of the European Union [29], in particular, those related to hydro-biological and hydro-morphological assessment of the water bodies and water management practices of Ukraine. In the last decade, several works had been published reporting the species diversity of the lakes in the forest zone of the Ukrainian Carpathians (Synevyr, Gropa, Maricheika, Hirske Oko and some others) and some information on the high-altitude lakes in the foothills of this mountain region [11,12,59,62–64]. Current intensified recreational pressure on the lakes and general hydrological changes in the region allow us to form a scientific working hypothesis assuming that the ecosystem of the lakes in the region is changing. The additional scientific question is whether the new results of ecological analysis based on a full taxonomic list of algae show any changes in comparison to previous results, focused on some groups of algae [61,65].

The importance of this study is supported by a threat to the existence of oligo-mesotrophic highland waterbodies and their peculiar species composition of aquatic organisms [11,66]. To confirm the hypothesis, the available data on algae species of the Nesamovyte Lake (Chornohora mountain group) were summarized from scientific papers covering over 100 years of research [2,3,63,67].

This study aims to document the algal and cyanobacterial composition of the Nesamovyte Lake using three datasets collected in the last hundred years and outlining the changes in the ecological state of the lake due to the bio-indication characteristics of the discovered species composition.

## 2. Materials and Methods

The Nesamovyte Lake is one of the highest mountain waterbodies of its type in Ukraine, located in the central, highest mountainous part of the Eastern (Ukrainian) Carpathians– Polonyno-Chornogora geomorphological area [68], belonging to Carpathian-Danube algofloristic sub-provinces according to algofloristic zoning of Ukraine [69]. The lake is located in the Carpathian National Nature Park (48°07′36.6′′ N, 24°32′26.4′′ E) and is situated in the glacial mountain trench on the eastern slope of Mount Turkul (Chornohora Ridge, Eastern Carpathians, and is approximately 5 km from the highest mountain in Ukraine—Hoverla) at an altitude of 1 748 m above sea level (a.s.l.) belongs to the water basin of the Prut River (Figure 1). According to the origin of the lake, it can be called a polymictic tarn that refers the following explanation 'a proglacial mountain lake, pond or pool, formed in a cirque excavated by a glacier' [70].

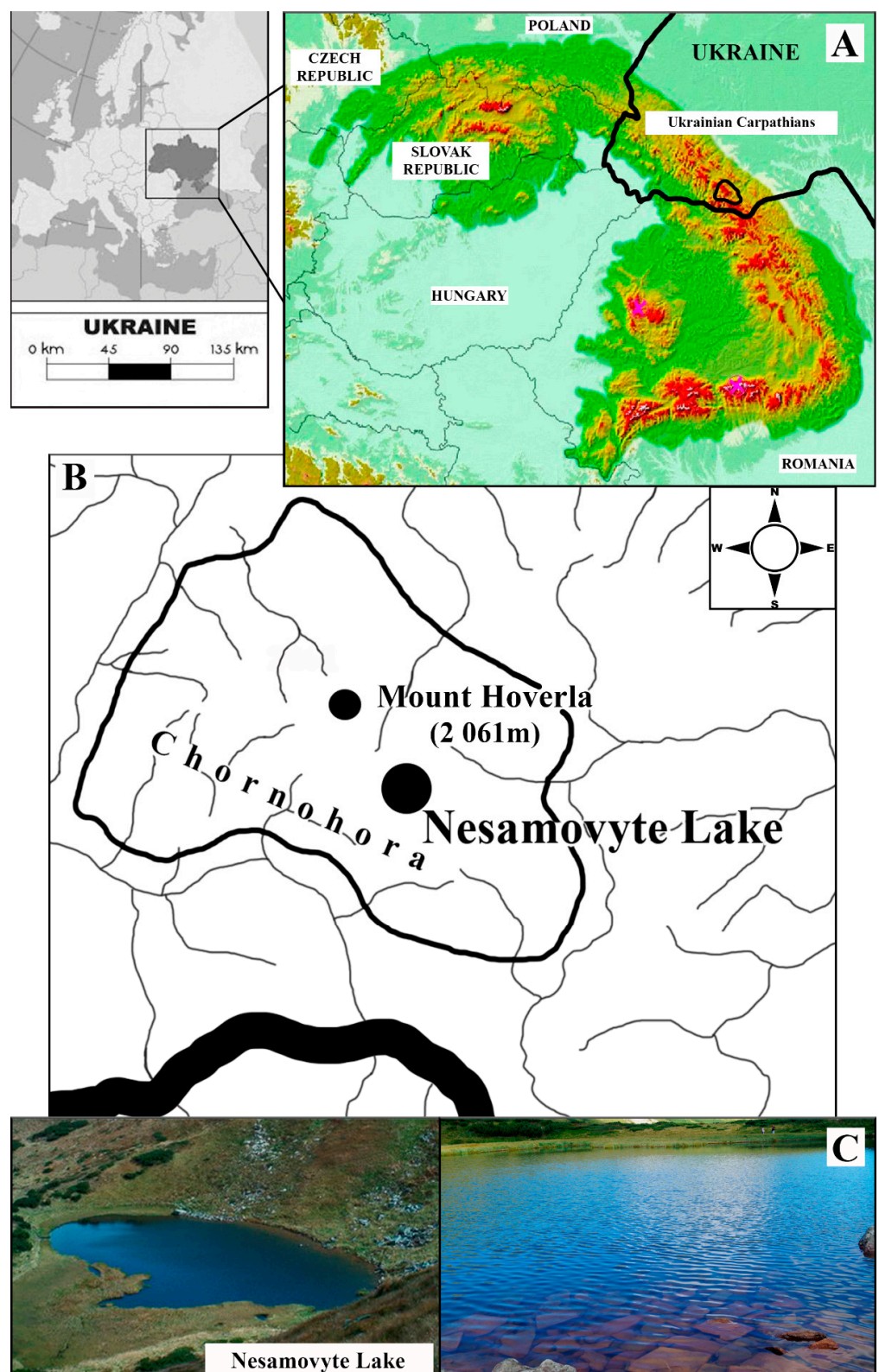

**Figure 1.** Map of the Carpathian region and the location of the Nesamovyte Lake. (**A**)—the Carpathian Mountains with its Eastern part the Ukrainian Carpathians. (**B**)—map-scheme of Chornohora Massif with the location of the lake. (**C**)—littoral zone of the lake.

The lake covers an area of about 0.4 hectares (about 88 × 45 m) and has a depth of about 2 m and, according to the EC Water Framework Directive [29], belongs to the

category of small and very small lakes. In general, the lake water is generated by rain and melting snow with a sandy muddy bottom (rocky from geological rocks of sandy flysch near the southern and western coast and sandy-muddy—in the northern part), it has a long period of freezing (October-May up to a depth of 1.5 m). It belongs to the cold climatic zone of the Carpathians. The summer water temperature normally is 18.3–(12.4) $\pm$3.2 °C. The lake has a slow underground runoff in the north-western direction, through a narrow channel on the alloy (swamp intrusion). This part of the lake is intensively overgrown by *Carex rostrata* Stokes and *Sphagnum* spp. Other moss-like organisms and sedge-sphagnum alloy cover the surface of the lake up to 35 cm from the shore and occupies about 40% of the lake bed area and reaches 1 m in thickness. The water in the lake corresponds now to the hydrocarbonate-sulphate-sodium class (Table 1) and belongs to zones of saprobity from oligo- to β-mesosaprobic [71–75]. The hydrochemical formula of water of the lake [76] is as follows: M0.018 SO4 HCO3 42/Na 79 Ca 21. According to EUNIS classification, [77], the Nesamovyte lake belongs to biotopes C1.1 or B1.1.1 (permanent oligotrophic lakes, ponds, puddles).

The material for this study was built on the analysis of the first record (1910–1920) for the Nesamovyte Lake published by Wołoszyńska, work with samples from 1967–1978 (collectors Prof. Z.I. Asaul and Prof. G.M. Palamar-Mordvintseva) and our previous investigations (2013–2018).

The species list provided by Wołoszyńska (collected in 1910) [67] was carefully analysed and describes the period between 1910–1920.

From 1967 to 1978, samples from Algoteca funds of M.G. Kholodny Institute of Botany of NAS of Ukraine (AKW)—NN 16855–16898 (samples from the year 1967–1978, collectors Prof. Z.I. Asaul and Prof. G.M. Palamar-Mordvintseva) were studied (44 samples). In addition, the data based on these samples published from the period of 1967–1978 [2–5] were included in the current analysis.

To characterize the modern period (2013–2021), 18 samples of net plankton (50–100 L of water filtered out), periphyton from vascular plants (dead and living parts of herbaceous plants (*Carex rostrata*) and pine branches (*Pinus mugo* Turra) and squeezes from the moss (*Sphagnum* spp.) were collected along the perimeter of the lake in August 2013–2018. To study the species composition of *Bacillariophyta*, the method of forming a combined sample from different studied substrates was used. This study is based on the living algal material from plankton (algal cultures, with the addition of BBM-medium) [78] and samples that were fixed with 4% formaldehyde solution.

The obtained results are comparable because almost identical methods for collecting and fixation of the algal material were used in all the studied periods (1910–1920, 1967–1978, 2013–2021).

The algae were studied and identified using light (LM) and scanning electron (SEM) microscopy, the permanent slides of diatoms were made according to the standard procedure using 35% $H_2O_2$ [79]. For LM investigations, the diatoms were fixed in the synthetic mounting medium Naphrax (refractive index 1.74) and investigated under a BX-53 microscope (Olympus, Tokyo, Japan). The slides are stored in the Algotheca of M.G. Kholodny Institute of Botany, NAS of Ukraine (AKW). For SEM analysis, the samples were put to specimen stubs, dried, covered with gold (10–20 nm) JFC-1600 sputter coater, and examined in a scanning electron microscope JSM-6060LA (JEOL, Tokyo, Japan) at the Institute of Botany in the Center of Electron Microscopy. The resulting micrographs were processed using the software packages Axiovision 4.3.7. (Carl Zeiss MicroImaging GmbH, Jena, Germany) and GIMP 2.8.10.m (Free Software Foundation, Inc., Boston, MA, USA).

Table 1. Hydrochemical characteristics of the Nesamovyte Lake [1].

| Variable | pH | $O_2$, mg $L^{-1}$ | Total Suspended Solids (TSS), mg $L^{-1}$ | Total Dissolved Solids (TDS), ppm | $Ca^{2+}$, mg $L^{-1}$ | $Na^+$, mg $L^{-1}$ | $HCO_3^-$, mg $L^{-1}$ | $SO_4^{2-}$, mg $L^{-1}$ | $Cl^-$, mg $L^{-1}$ | $NO_3$, mg $L^{-1}$ | Cd, µg $L^{-1}$ | Cr, µg $L^{-1}$ | $Pb^{2+}$, µg $L^{-1}$ |
|---|---|---|---|---|---|---|---|---|---|---|---|---|---|
| Value | 6.2–6.4 ± 0.4 | 7–10.7 | 1.6 ± 0.4 | 8.2–9.8 (12–98.6) | 1.30 ± 0.3 | 1.93 ± 0.24 | 3.66 ± 0.145 | 3.98–15.0 | 0.71–1.42 | 2.0 ± 0.4 | 0.26 | 2.37 | 0.39 |

[1] according to [66,70,72] and our research.

Identification of the species diversity was carried out according the Süßwasserflora von Mitteleuropa [80–85] with some newer updates from Diatoms of Europe [86–89], Diatomeen im Süßwasser-Benthos von Mitteleuropa [90] and Flora of algae of Ukraine [91–97] as well as with updates from [98–102], and electronic resources [103]. The identified taxa, as well as all algal species lists from previous years of studies of the territory were validated using the AlgaeBase system [104] and "Algae of Ukraine . . . " [105] monographic series.

Ecological bioindicator species analysis was based both on the historical data (1910–1920 and 1967–1978 collections) and the results of our studies (2013–2021) [2–4,61,63,65,67, 106,107]. The following ecological characteristics were used: Habitat preference, streaming and oxygenation, pH [108], salinity, trophic state and class of organic pollution [109–112]. Identification of organic pollution by the values of saprobity indices and indicator groups, equated to water quality classes [113,114]. Considering the groups of indicators that we identified, the intervals of the quality classes were distributed as follows: I—0–0.5 (x, x-o); II—0.6–1.5 (o-x, x-b, o, o-b), III—(b-o, o-a, b, b-a) and IV—(a) [112,115–117]. Ecological features of the species were presented according to [112,117,118].

The Cluster analysis of algal composition was carried out using the Paleontological Statistics Software (PAST, Palaeontological Association, Hammer & Harper, Oslo, Denmark, Galway, Norway, Denmark, Ireland) to measure the degree to which species composition was similar among the studied periods (1910–1920, 1967–1978, 2013–2021) [119]. For this purpose, the presence/absence of data were used in the meaning of the Sørencen coefficient, calculated in the program as Bray-Curtis Similarity index [120].

## 3. Results

### 3.1. Species Composition of Algae during 100-Year Investigated Period

The taxonomic structure of the algal composition of the Nesamovyte Lake is a summary of the available data on the species composition of algae for 100 years of investigations during the following study periods: 1910–1920, 1967–1978 to 2013–2021 [2,4,61,63,65,67,106,121]. It is represented by 234 species (245 infraspecific taxa (inft) composing eight divisions, 15 classes, 33 orders, 55 families, 100 genera of Cyanobacteria and algae (Table 2, Figure 2).

**Table 2.** Systematical composition of algae of the Nesamovyte Lake (1910–2021).

| Division | 1910–1920 | | | 1967–1978 | | | 2013–2021 | | | 1910–2021 | | |
|---|---|---|---|---|---|---|---|---|---|---|---|---|
| | Genera | Sp (inft) | % | Genera | Sp (inft) | % | Genera | Sp (inft) | % | Genera | Sp (inft) | % |
| Cyanobacteria | 2 | 2 | 3.0 | – | – | – | 3 | 3 | 1.8 | 5 | 5 | 2.1 |
| Euglenozoa | – | – | – | 6 | 10 | 14.6 | 5 | 8 | 4.8 | 8 | 12 | 5.1 |
| Ochrophyta | 1 | 1 | 1.5 | 2 | 3 | 4.9 | 1 | 1 | 0.6 | 3 | 4 | 1.7 |
| Cryptophyta | – | – | – | 1 | 1 | 2.4 | 2 | 2 | 1.2 | 2 | 2 | 0.9 |
| Bacillariophyta | 7 | 15 (16) | 23.4 | – | – | – | 44 | 115 (117) | 69.7 | 46 | 122 (125) | 52.1 |
| Miozoa | 2 | 2 | 3.0 | – | – | – | – | – | – | 2 | 2 | 0.9 |
| Chlorophyta | 4 | 4 | 6.0 | – | – | – | 11 | 12 | 7.3 | 13 | 14 | 6.0 |
| Charophyta | 16 | 43 (45) | 64.1 | 8 | 27 (31) | 58.5 | 14 | 24 (25) | 14.6 | 21 | 73 (81) | 31.2 |
| Total | 32 | 67 (70) | 100 | 17 | 41 (45) | 100 | 80 | 165 (168) | 100 | 100 | 234 (245) | 100 |

The basis of this taxonomic and species diversity for the lake is formed by Bacillariophyta and Charophyta, uniting together giving about 57% generic diversity and over 83% species and infraspecific composition of algae in different habitats of the Nesamovyte Lake. Less diverse, but indicative for presenting the dynamics of changes in taxonomic structure over time, presented the divisions of Chlorophyta (6.0%) and Euglenozoa (5.1%). A slight variety of species is inherent for the divisions' Cyanobacteria and Ochrophyta (Chrysophyceae) (2.1–1.7%), as for Cryptophyta and Miozoa (Dinophyta)—they are noted as having a few representatives (>1.0%).

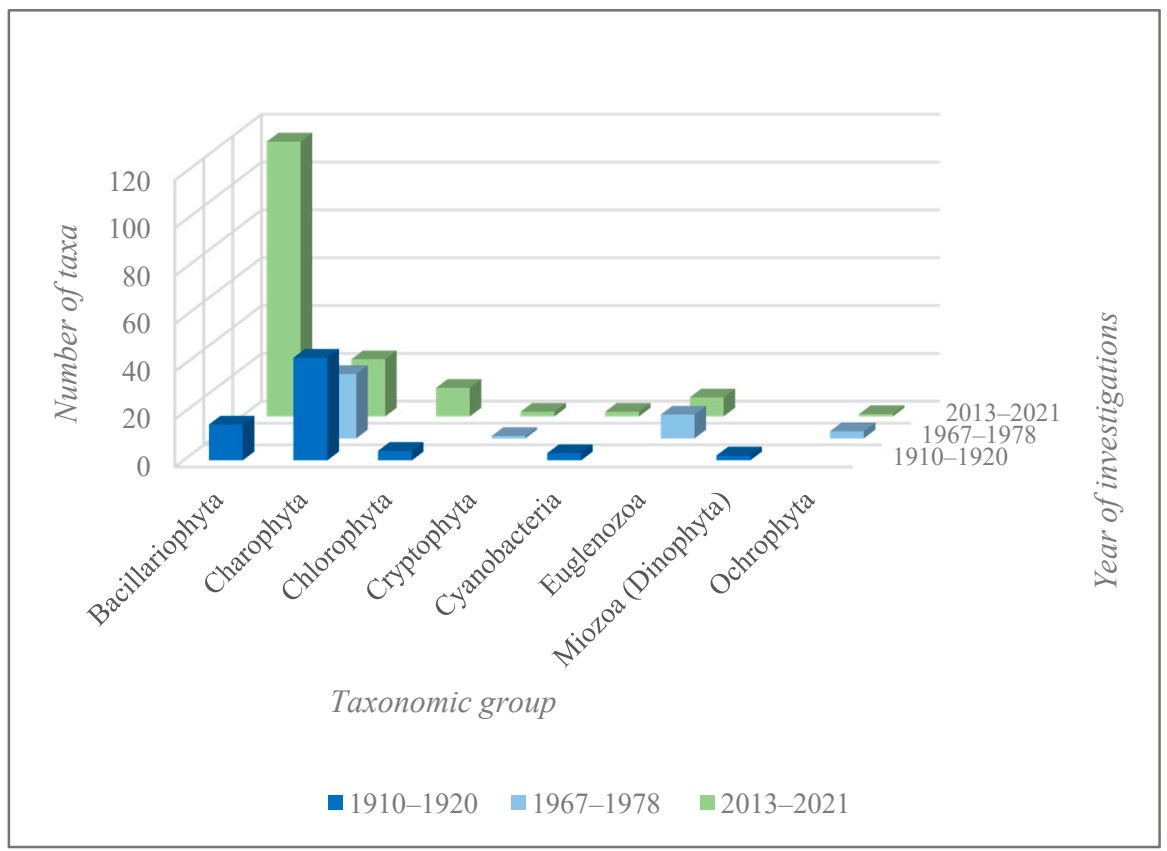

**Figure 2.** Species composition of algae and cyanobacteria in the Nesamovyte Lake (split by taxonomic groups and three stages of research in 1910–2021).

At the same time, the leading set of orders of these taxonomic groups is formed by representatives of Desmidiales (15 genera—66 species—75 inft), Naviculales (10 genera—50 species), Cymbellales (5–25), Achnanthales (5–11), Eunotiales (1–11), Sphaeropleales (6–7), Fragillariales (4–9), Zygnematales (6–6) and Euglenidida (2–5). These orders account for around 56% of the generic and species composition of algae in the Nesamovyte Lake. Leading genera of the algal composition are *Staurastrum* Meyen ex Ralfs (21–22 inft), *Pinnularia* Ehrenberg (14–16), *Euastrum* Ehrenberg ex Ralfs (12–15), *Navicula* Bory (12), *Cosmarium* Corda ex Ralfs (12), *Eunotia* Ehrenberg (11), *Gomphonema* Ehrenberg (10), *Encyonema* Kützing (5), *Planothidium* Round & L. Bukhtiyarova (5) and *Closterium* Nitzsch ex Ralfs (4). The named 10 genera cover about half (45.5%) of the whole diversity of species composition of this waterbody. However, almost 60% of genera are represented only by one species, which could serve as a characteristic feature of the studied lake flora.

*3.2. Comparison between Studied Periods of Species Composition of Algae during 100-Year Investigated Period*

Comparative floristic analyses for three periods of investigations showing the similarity of algae composition is presented in Figure 3. For building the graph, the species lists for the periods 1910–1920, 1967–1978 and 2013–2021 were compared. The graph demonstrates a low similarity between the studied periods, however, some higher similarity for 2013–2021 and 1910–1920 can be noted in contrast to the middle period of the investigations.

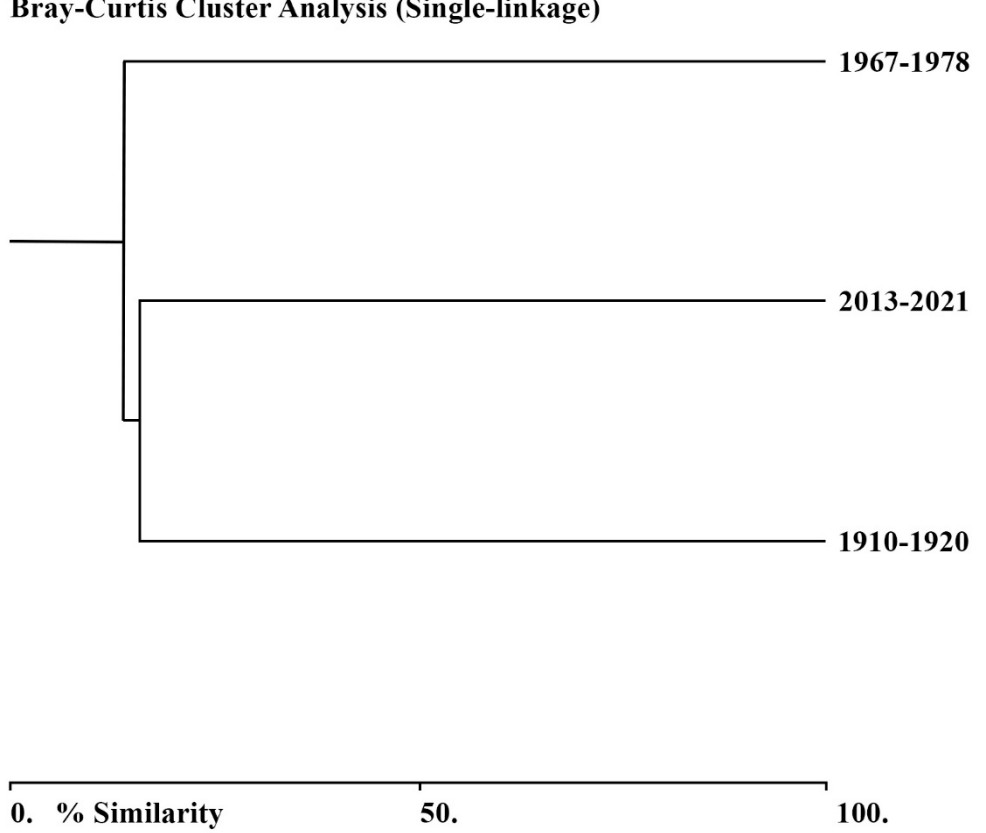

**Figure 3.** Bray-Curtis tree of species composition comparison in three periods of the Nesamovyte Lake study (1910–1920; 1967–1978; 2013–2021).

The first studies of the species composition of algae in this lake were conducted in the early twentieth century by Professor J. Wołoszyńska (based on the 1910 samples of Prof. M. Raciborski) [67]. With the help of this study, the presence of only 67 species (70 inft), in particular Bacillariophyta (15–16 inft), Charophyta (43–45 inft), Chlorophyta (4), Miozoa (Dinophyta) (2), Ochrophyta (Chrysophyceae) (1) and Cyanobacteria (2) were noted.

Leading taxonomic groups by species richness were charophytes (desmids—64.1%) and diatoms (23.4%), combining more than 87% of the whole composition of algae of this waterbody. At the same time, the author noted that diatoms do not make a "*significant contribution to the diversity of species composition of the lake*" ([67], p. 144) (probably meaning a much lower percentage of diatoms). Some distinguishing features for the uniqueness of the diatom composition at the genera level—*Eunotia*, *Pinnularia* and *Neidium* Pfitzer— were underlined.

Half a century later (1967–1978 studied period) the targeted studies of species diversity of euglenoid and desmids algae [2–4] from different ecotopes of the lake revealed 102 species of algae. The core divisions of Charophyta (66) and Euglenozoa (9), were added by different divisions presented by Cyanobacteria (3), Cryptophyta (1), Ochrophyta (2), Miozoa (2), Chlorophyta (4) and Bacillariophyta (15) (Table 2).

Our investigations of the modern composition of algae in the Nesamovyte Lake (2013–2021) confirm its species diversity (165 species—168 inft, belonging to 80 genera, 47 families, 28 orders, 11 classes and seven divisions—Table 2) and provide insights into the taxonomic composition and leading complexes of the species. According to the results of this study, the species composition was presented by Bacillariophyta (115 species—117 inft, namely ~70% of total species composition), Charophyta (24–25, ~15%) and Chlorophyta (12, ~7.3%), sparse—Euglenozoa (5–8, ~5.0%) and the lower—Cyanobacteria (3, ~1.8%) and Cryptophyta (2 > 1%).

The basis of the species composition of Bacillariophyta is formed by the following orders: *Naviculales* (38.8%), *Cymbellales* (21.5%), *Achnanthales* (10.3%), *Eunotiales* (8.6%), *Fragilariales* (5.2%), and among families—*Naviculaceae* (12.1%), *Cymbellaceae* (12.1%), *Pinnulariaceae* (11.3%), *Gomphonemataceae* (9.5%), *Eunotiaceae* (8.6%), *Achnanthaceae* (7.7%) [61,65,121]. These families comprise more than 61% of the species composition of diatoms in the Nesamovyte Lake. The genera characterized by high species diversity are the following: *Pinnularia* (12 species (14 inft)—12.1%), *Navicula* (11 species—9.5%), *Eunotia* (10 species—8.6%), *Gomphonema* (10 species—8.6%), *Encyonema* (5 species—4.3%). The complex of leading species of diatoms from different ecotopes (mainly plankton) of the lake with high quantitative indicators of development was formed by *Tabellaria flocculosa* (Roth) Kützing, *Eunotia minor* (Kützing) Grunow and *Frustulia crassinervia* (Brébisson) Lange-Bertalot et Krammer. Their abundance varied from 4 to 5 according to Starmach scale [122]. In addition, the regionally rare species of the flora of Ukraine, which are known to be from this lake were revealed—*Cymbella lange-bertalotii* Krammer, *Encyonema neogracile* Krammer, *Eunotia tetraodon* Ehrenberg, *Pinnularia macilenta* Ehrenberg, *P. subanglica* Krammer (Figure 4), *Skabitchewskia peragalloi* (Brun & Héribaud) Kuliskovskiy & Lange-Bertalot, and *Pinnularia falaiseana* Krammer. Some of them have a pronounced disjunctive distribution in the world and are considered as rare species [65].

A high variety of modern species composition found for Bacillariophyta [65] contrasts sharply with the data at the beginning of the XXth century [67], when only 16 species were reported (17 inft). Moreover, for the modern period of studies, the presence of arctic diatom species that also are regionally rare for Ukraine [105], and in particular, *Cavinula pseudoscutiformis* (Hustedt) D.G. Mann & Stickle, *Pinnularia rhombarea* Krammer, *P. rupestris* Hantzsch, *P. subanglica* Krammer were found (Figure 4).

According to the results of the "green" phyla algae flora (Charophyta and Chlorophyta), the comparison modern and studied periods of 50 (1967–1978) and 100 years [3,4,67] were conducted. A decrease in the species diversity of Charophyta (from 43 species (45 inft) in 1910–1920 to 27 (31)—in 1967–1978 and up to 24 (25) nowadays) and increase of Chlorophyta (four species—13 species, correspondingly for 1910–2021) were noticed [61,121]. Representatives of the classes Zygnemaphyceae (14.6%) and Chlorophyceae (7.3%) comprise about one-fifth of all the algal species in the Nesamovyte Lake nowadays. A significant role in forming this diversity belongs to the representatives of the order Desmidiales (11.6%), and species diversity of *Zygnematales* and *Sphaeropleales*—low and is between 3.7% and 3.0%, correspondingly. Among genera by species diversity differ *Euastrum* Ehrenberg ex Ralfs and *Staurodesmus* Teiling ex Compere (five species—3.0%, each of them). Eighteen (18) genera are represented only by one species each. In turn, compared to the beginning of the XXth century, the presence of species of genera has not been confirmed for *Actinotaenium* (Nägeli) Teiling, *Cylindrocystis* Meneghini ex De Bary, *Micrasterias* Meneghini ex De Bary, *Netrium* (Nägeli) Itzigsohn & Rothe, *Teilingia* Bourrelly and *Coelastrum* Nägeli, and for half a century (1967–1978)—species of genera *Penium* Brébisson ex Ralfs, *Sphaerozosma* Corda ex Ralfs, *Spirotaenia* Brébisson and *Tortitaenia* Brook.

At the same time, rare species of genera have now been identified: *Hyalotheca* Ehrenberg ex Ralfs, *Euastrum* Ehrenberg ex Ralfs and *Tetmemorus* Ralfs ex Ralfs. These species are inherent to the flora of water bodies in mountainous regions in general [34]. During the current research period (2013–2021), the presence of rare coccoid green algae (in particular, *Pediastrum braunii* Wartmann = *P. tricornutum* Borge var. *alpinum* Schmidle) had also been unconfirmed. In addition to rare forms of these three genera, the following algae species were found now for Nesamovyte lake the first time: representatives of filamentous charophyte algae and mucilage-forming green coccoid, flagellar algae and cyanobacteria (*Mougeotia* C. Agardh, *Spirogyra* Link, *Zygnema* C. Agardh, *Oedogonium* Link ex Hirn, *Botryococcus* Kützing, *Chlamydomonas* Ehrenberg, *Mucidosphaerium* C. Bock, Pröschold & Krienitz, *Chlorella* Beijerinck, *Mychonastes* P.D. Simpson & S.D. Van Valkenburg, *Westella* De Wildeman, *Anabaena* Bory ex Bornet & Flahault).

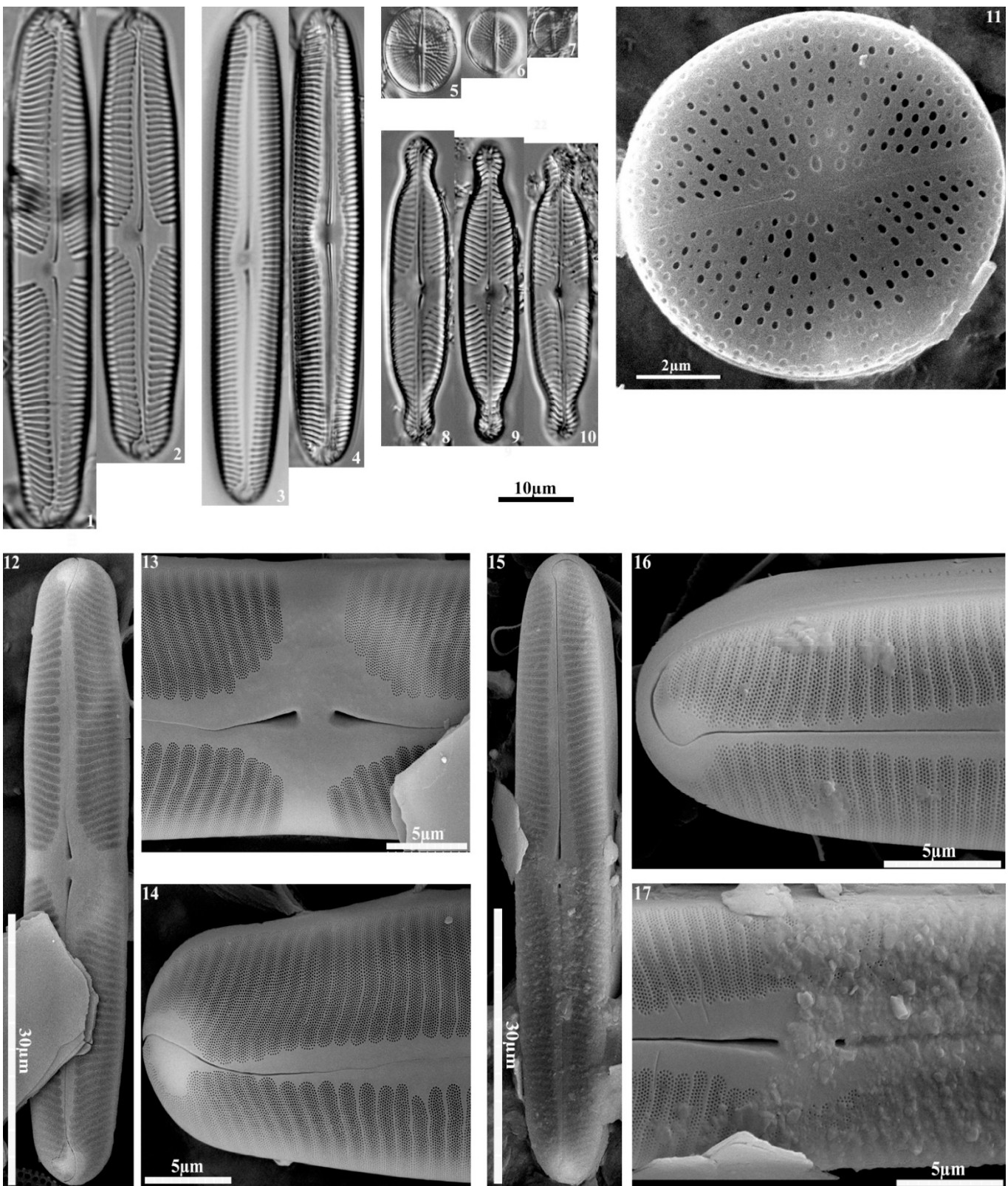

**Figure 4.** Regionally rare diatom species from the Nesamovyte Lake: 1–2, 12–14—*Pinnularia rhombarea,* 3–4, 15–17—*P. rupestris*, 5–7, 11—*Cavinula pseudoscutiformis,* 8–10—*P. subanglica*.

According to the results of comparative analysis of species composition of Charophyta division in the Nesamovyte Lake for 100 years [67], a change in the complex of leading groups of Desmidiales/Zygnematales according to quantitative characteristics (frequency of occurrence) was observed. The communities *Cylindrocystis brebissonii* (Ralfs) De Bary, *Actinotaenium cucurbita* (Brébisson ex Ralfs) Teiling ex Růžička, *Cosmarium staurastriforme* Gutwinski, *C. venustum* (Brébisson) W. Archer var. *excavatum* (Eichler et Gutwinski) West et G.S. West, *Euastrum insigne* Hassall ex Ralfs, *E. humerosum* Ralfs var. *humerosum* and var. *subintermedium* Schröder, *E. didelta* (Turpin) Ralfs, *Staurastrum muricatiforme* Schmidle in 1910–1920, have changed to the similar *Staurastrum senarium* Ralfs f. *senarium* and f. *tatricum* Raciborski, *Euastrum pinnatum* Ralfs, *E. humerosum* var. *humerosum* and var. *affine* (Ralfs) Raciborski, *E. didelta* Ralfs, *E. amoenum* F. Gay in 60–70-es of XX[3]. The modern grouping of these algae has happened in the beginning of the XXIst century and the community was formed by *Hyalotheca dissiliens* Brébisson ex Ralfs, *Netrium digitus* (Ehrenberg ex Ralfs) Itzigsohn et Rothe emend. Ohtani, *Euastrum humerosum* var. affine, *E. ansatum* Ehrenberg ex Ralfs and *Staurastrum polytrichum* (Perty) Rabenhorst.

The comparison between the modern period and total list of algae showed that the species diversity of algae nowadays makes up over 70% of the total number of the found species and is characterized by an increase in the number of widespread forms. At present, the basis of the taxonomic and species diversity of the lake is being formed by Bacillariophyta and Charophyta, which in total exceeds about 57% of the genera amount and over 84% of the species and infraspecific composition of algae. Less diversely represented are Chlorophyta and Euglenozoa (>6.0%), and Cyanobacteria and Ochrophyta—are quite low (>2.1%). The dynamics of change in the composition of the algae over the period of 100-years shows an increase of Bacillariophyta (from ~23% to above 70%) and the change in the leading taxonomic group—Charophyta.

The appearance and floristic significance of Euglenozoa (~5.0%) over the last half-century—as one of the indicators to the increased degree of the trophic state of the waterbody—was also recorded. In turn, the preservation of secondary importance in terms of species composition groups as Cyanobacteria, Ochrophyta and Cryptophyta were noted. Representatives of Miozoa (Dinophyta) according to comparison with the latest investigations were not identified. In addition, the presence of widespread species of filamentous charophytes and mucilage forming green coccoid and filamentous algae has been noted (*Mougeotia*, *Spirogyra*, *Zygnema*, *Oedogonium*, *Botryococcus*, *Chlamydomonas*, *Mucidosphaerium*, *Chlorella*, *Mychonastes*). Also, some members of Euglenozoa group were found [61].

The "blooming" in the water of the Nesamovyte lake was also tested. It is assumed, that the blooming is caused by the mass development of green colonial coccoid algae *Botryococcus terribilis* Komarek et Marvan (Trebouxiales, Trebouxiophyceae), and which was noted for the first time during the summer of 2015 [106,123].

Our previous investigations [61,63,65] for the comparison of the floral community similar to the Chornogora mountain group revealed a low level of similarity and spatial separation among the lakes of the group (Figure 5). However, the Jaccard coefficient with 43% distinguished a group of lakes—Nesamovyte, Bolotne Oko and Tsyclop with similar algal composition.

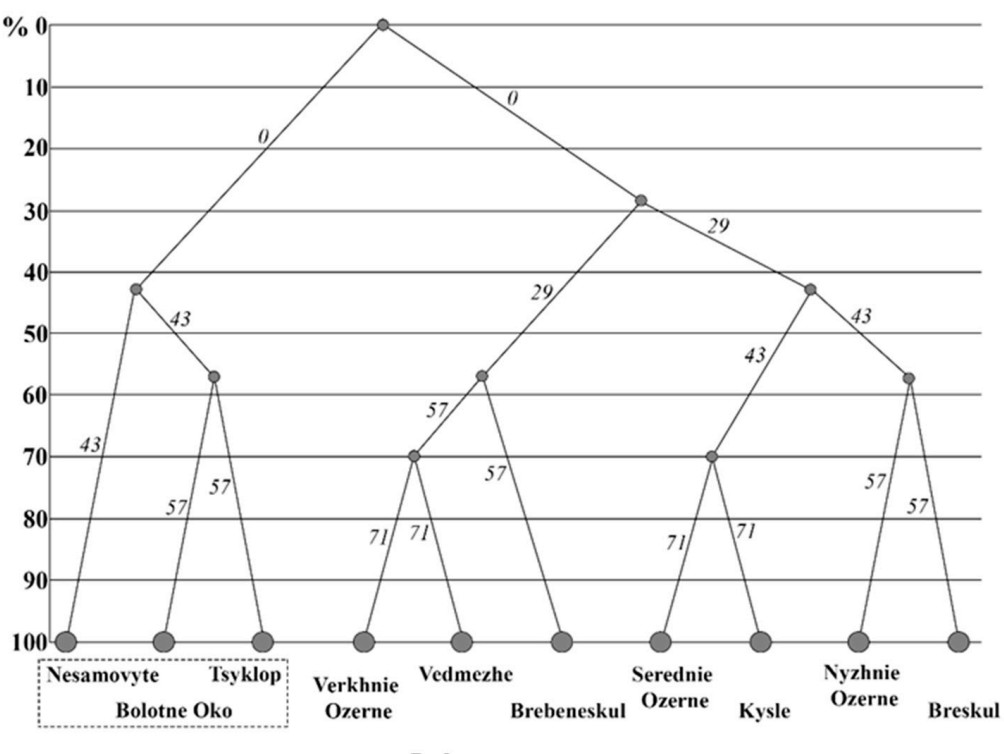

**Figure 5.** Dendrogram of floristic similarity of species composition of algae in the Nesamovyte Lake with regionally close lakes of the Chornogora mountain group (Jaccard coefficient).

### 3.3. Ecological Characteristics of the Nesamovyte Lake Due to Algal Preferences

Environmental analysis is based on the types of indicators, which are grouped according to the following characteristics: habitat preferences, streaming and oxygenation, pH, salinity, trophic state, organic pollution (water quality class) (Figure 6). With the help of ecological preferences of species grouped by the previously mentioned time intervals (I—1910–1920; II—1967–1978; III—2013–2021), the ecological characteristics of these periods are also presented.

Characteristics of habitat preferences are determined on a base of indicator species (according to [112]). From 1910 to 1920, species preferring the attached to the substrate way of existence, were formed by benthic species (B) that amounted to 19 taxa or 43.2% (from the total number of indicators of habitat preference) and plankto-benthic (P–B)—15 taxa or 34.1%. For 1967–1978, 17 benthic (B) taxa dominated, composing 53.1% of the total amount of indicators of habitat preferences. However, the amount of plankto-benthic (P–B) taxa were less—eight taxa or 25%. During the modern period (2013–2021), the benthic algae (B) prevailed composing 61 taxa or 50% from total indicator number of habitat preferences, also plankto-benthic (P–B)—47 taxa or 38.5%. Over the studied periods, the restructuring of dominant groups of habitat preference indicators were reported.

Streaming and oxygenation analysis for 1910–1920 revealed the indicators of medium-mobile waters, medium enriched with oxygen or standing-streaming (st-str) composing 10 taxa or 45.5% from the total number of indicators of this ecological preference. Somewhat less amount (seven taxa or 31.8%) was formed by aerophytic forms (ae) that are also called pseudoaerial species. For 1967–1978, indicators of standing-streaming (st-str) prevailed and their amount was 53.8% or seven taxa. Analysing the modern period (2013–2021), indicators of standing-streaming (st-str) waters, also prevailed and their amount equalled 58 taxa-indicators (conjugates, diatoms and green algae), composing 58% from the total amount of taxa of indicators of this group.

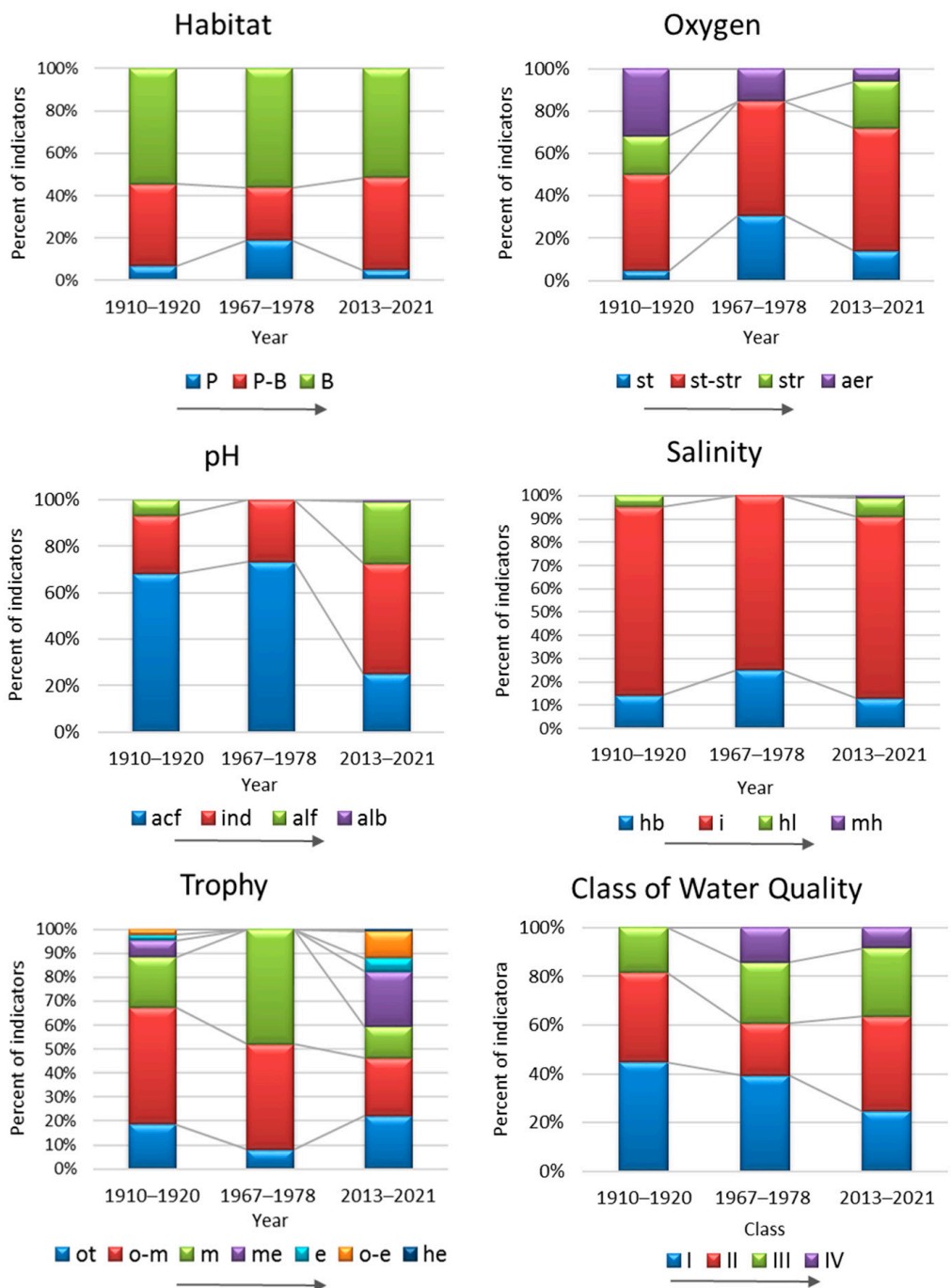

**Figure 6.** Bioindication plots for ecological analysis of *Habitat preference* (Habitat): P—planktonic, P-B—plankto-benthic, B—benthic; *Streaming and oxygenation* (Oxygen): st—organisms that favour standing water with low oxygenation, st-str—organisms that favour low-flow, moderately oxygenated water; str—organisms that favour streaming water with a high level of oxygenation, aer—aerophyles; *pH*: acf—acidophiles, ind—pH-indifferents; alf—alkaliphiles; alb—alkalibiontes; *Salinity*: hb—halophobes, i—indifferents, hl—halophiles, mh—mesohalobes, *Trophic state* (Trophy): ot—oligotraphentes, o-m—oligo-mesotraphentes, m—mesotraphentes; me—meso-eutraphentes; e—eutraphentes; o-e—from oligo- to eutraphentes; he—hypereutraphentes and *Class of water quality* (according to organic pollution) in the Nesamovyte Lake (for the periods: I—1910–1920, II—1967–1978, III—2013–2021.

Indication of pH (according to [108,112]) for 1910–1920 revealed the prevalence of acidophils, which included 30 taxa, composing 68.2% of the total amount of indicators of

this group. For 1967–1978, acidophils also prevailed containing 22 taxa, that was equal to 73.3% from the total amount of the taxa. For modern period (after 2013), the indifferents (ind) prevailed and their amount equalled 50 taxa or 43.1% with the relatively large number of alkaliphiles (alf), composing 31 taxa or 26.7%. Noteworthy is the number of acidophilic (acf) species that is formed by 29 taxa or 25% of the total number of indicators of this group for this period of investigations.

The salinity indicators (according to [109,112]) covered from 8.9% to 60.5% of the total amount of species for each studied period. During all the years of investigations, the significant prevailing of indifferents (i) was recorded.

The trophic state of the Nesamovyte Lake in 1910–1920 indicated prevailing of oligo-mesotraphentic (o-m) indicators that covered 21 taxa or 48.8% from the total amount of taxa for this year. In 1967–1978, the mesotraphentic (m) indicator taxa, which numbered 12 taxa or 48% along with oligo-mesotraphentic (o-m) composing 11 taxa or 44% prevailed. During the modern period (2013–2021), the following groups formed majority of indicators: oligo-mesotraphentic (o-m): 26 taxa or 24.1%, meso-eutraphentic (me): 25 taxa or 23.1% and oligotraphentic (ot): 24 taxa or 22.2%. It is also worth noting the presence of hypereutraphetic (he) species as well as an increase in the number of eutraphetic (e) species for this period that was not recorded from the earliest periods.

Characterization of organic pollution (saprobity) for 1910–1920, indicated class I water quality formed by 17 indicator taxa composing 44.7% from all groups of indicators for this period. In 1967–1978, indicators of class I predominated (11 taxa or 39.3% of indicators of characterizing parameter of this year). However, the presence of indicators of class IV water quality was also noticed. During the modern period, indicators of class II of water quality (46 taxa or 39% of indicators characterizing this parameter for this year) prevailed with the presence of indicators of the class IV quality.

## 4. Discussion

### 4.1. The Comparison of the Nesamovyte Lake Diversity with Similar Lakes in Adjacent Areas

A comprehensive study of the entire species composition of algae revealed the high diversity and richness in the algae of Lake Nesamovyte, its specificity and uniqueness as well as the need for conservation and protection. The data on similar lakes of the Carpathian-Tatra region (including the Western Carpathians) covering different taxonomic groups of algae and the dynamic changes in their species composition over a long period of studies or even under modern conditions are limited. Tatra National Park, Slovakia [47–49] and Tatrzański Park Narodowy, Poland [35,37,46] are regionally close regions with available taxonomical characteristics of diversity of all divisions of algae. However, these studies differ in the affiliation of the studied lakes to other climatic zones of mountain groups, excellent altitude gradient location and hydrology of lakes and the period of the study, which makes it impossible to compare correctly the available results. Besides, a variety of studies of mountain lakes in this region focus on the study of general hydrobiological features of these reservoirs, and in terms of diversity—aimed at the careful analysis of the species composition of only one taxonomic group of algae of modern and fossil algal composition of adjacent areas or one ecotope of mountain reservoirs [8,14,52–55,57,124,125]. This purposeful nature of the taxonomic study also complicates the comparative aspect of studying the diversity and specific features of the species composition of the algae flora of the Nesamovite Lake.

### 4.2. Comparison between Studied Periods of Species Composition of Algae during the 100-Year Investigated Period

The total dataset regarding the composition of algae covering a 100-year period of investigations presents unique information for a comparative study between the different decades of studies and reveals some defining features in each. The low similarity between years of studies could be a result of the changes that happened with the Nesamovyte lake. Ecological analysis allows us to determine possible reasons for such changes. Some similar-

ity between 2013–2021 and 1910–1920, is in contrast to the 1967–1978 years of investigation (Figure 3) and is attributable to the nature of the study of species diversity during the middle period. The coverage of only two taxonomic groups of algae—Euglenozoa [2] and Desmidiales [3] were conducted during 1967–1978, while the studies of the species diversity at the beginning of XXth century and XXIst centuries covered the diversity of all taxonomic groups of algae.

The species composition of the lake during 1910–1920 revealed the presence of specific 27 alpine Holarctic species of desmids and coccoid green algae. It also confirmed uniqueness as well as some affinity to the "Montane-Alpine" component of the species composition of the studied "flora" of the Eastern Carpathians with similar lakes of the Tatra, Sudetes and Alps. At the same time, the author provides the common species lists of algae with the lakes of Tatra, Sudetes and Alps (*Cylindrocystis brebissonii* (Ralfs) De Bary, *Penium cylindrus* Brébisson ex Ralfs (=*Penium cylindrus* var. *subtruncatum* Schmidle), *Tetmemorus laevis* Ralfs ex Ralfs (=*T. laevis* var. *ornatus* Schmidle), *Tetmemorus brebissonii* Ralfs var. *minor* De Bary, *Actinotaenium cuccurbita* (Brébisson ex Ralfs) Teiling ex Růžička (=*Cosmarium cucurbita* Brébisson var. *cucurbita* et var. *attenuatum* G.S. West), *Euastrum montanum* West et G.S. West (=*Cosmarium subreinschii* Schmidle var. *boldtianum* Schmidle), *Euastrum insigne* Hassall ex Ralfs, *Staurastrum scabrum* Brébisson, *S. muricatiforme* Schmidle and other) and alpine holarctic species of charophytes (desmids) and green (coccoid) algae—*Cosmarium nasutum* Nordstedt f. *tatrica* Gutwinski, *C. staurastriforme* Gutwinski, *C. polonicum* Raciborski (=*C. vogesiacum* Lemaire), *Euastrum aboense* Elfving, *E. binale* (Turpin) Ralfs var. *papilliferum* Gutwinski, *Pediastrum tricornutum* Borge var. *alpinum* Schmidle, *Teilingia granulata* (J.Roy & Bisset) Bourrelly (=*Sphaerozosma granulatum* Roy et Bisset var. *trigranulatum* West), *Staurastrum subavicula* (West) West et G.S. West f. *tyrolense* (Schmidle) G.W.Prescott, C.E.M.Bicudo & W.C.Vinyard (=*Staurastrum vastum* Schmidle var. *tyrolense* Schmidle), *Tortitaenia alpina* (Schmidle) Brook [67] (p. 147).

The studies during 1967–1978 [2–4] showed certain changes in the taxonomic structure of the "flora" and the importance of the following groups: euglenoid algae in different ecotopes of the lake as well as the distinctive high diversity of desmids [3,61].

The modern species composition of algae in the Nesamovyte Lake (2013–2021) has extremely high rates of species diversity noted for Bacillariophyta. This fact makes the modern period significantly different from the two previous periods of studying the floristic composition of algae during 1910–1920 [67] and 1967–1978 [2–4], i.e., 100 and 50 years ago respectively [11]. However, this fact can probably be explained by the current purposeful study of the species composition of diatoms and apparently the influence of a set of ecological and geographical reasons for such transformation, in particular the possible settlement of various ecotopes of the lake by the widespread forms as well as mesotrophic and eutrophic species of the plains area. The reason for the detected diversity of diatoms can possibly be explained by the change in the hydrochemical parameters of the lake. Comparisons by our study with the previous studies as well as inter-comparison of the previous studies revealed more than an 8-fold increase in TDS level over the past 50 years (12–98.6 mg L$^{-1}$) and a 4-fold increase in the number of sulfates (up to 15.0 mg L$^{-1}$). Species of this taxonomic group were found to be resistant to heavy metal ions Cd—0.26 mg L$^{-1}$, Cr—2.37 mg L$^{-1}$, Pb—0.39 mg L$^{-1}$ [70,72,74]. Furthermore, high values of the diversity of Bacillariophyta as the leading taxonomic group is distinctive of the highlands of Europe [14,26,52,54,55,60] and correspond to our obtained results. Besides, oligotrophic lakes of the alpine zone are refugium for the conservation of rare and conditionally endemic species [15,39,126], and therefore the presence of European and regionally rare species of Bacillariophyta in the ecotopes of the Nesamovyte Lake is an additional argument for the protection and preservation of the ecosystem of this lake and its diversity as a habitat for their existence.

Materials used in our study for the diversity of the "green" phyla algal flora (Charophyta and Chlorophyta) of this lake during the last decade of the XXI confirm the richness of their species composition and floristic importance in the ecosystem of the lake (Table 2)

compared to the period of 50 (1967–1978) and 100 years [3,4,67]. The species composition of Chlorophyta in alpine lakes is characterized by a low level of diversity in general, but often reveals features of uniqueness and ecological specificity [34,49,58,127], which was reported at the beginning of the twentieth century at the first stage of investigations of algae in oligotrophic lake Nesamovyte [61,67]. The findings of noted algae from "green" phyla indicate the probable degradation of the alpine oligotrophic ecosystem of the lake with the earliest possible data (early twentieth century), its transformation to oligo-mesotrophic status nowadays and the settlement of its ecotopes by common species with a wide ecological amplitude [11,61,66,121].

In addition, the comparison of the species diversity of algae in the beginning of XXwith a modern period revealed changes in the "Montane" complex. This resulted in the disappearance of the majority (over 80%) of the taxa of the "Montane-Alpine" complex that is common for high-altitude lakes of the Alpine-Carpathian region and suggested the mentioned above degradation of the ecosystem. Besides, this fact once again proved the loss of the alpine oligotrophic indicator species [15,126]. However, the presence of a unique algal composition typical for alpine lakes in Europe—a complex of conditionally arctic algae species, including diatoms was noted [26,39,67,121].

The change in ecological conditions became first evident in the 1960s during focused investigations of different ecotopes of the lake for Euglenozoa and Cyanobacteria [2,61,121]. These groups were considered as the typical representatives of the flora in alpine lakes of Europe and indicated a change of the ecosystem of lakes with high levels of organic pollution. This fact confirms a probable increase of trophic state in the Nesamovyte Lake [124]. Such evidence may indicate a degradation of the ecosystem of the high-altitude oligotrophic lake, its transformation to the mesotrophic type and its settlement by representatives with a wide ecological amplitude that started somewhere in between the two studied periods from 1910–1920 to 1967–1978 [61]. And nowadays, the confirmation of such changes in the ecosystem, its violation and increase in trophic state is in our opinion supported by the fact of "blooming" of the lake with green colonial coccoid algae *Botryococcus terribilis*, which was previously noted only in mesotrophic lakes in Australia [128].

An analysis of geographically close and hydrologically related lakes of the Chornogora mountain group (Figure 5) testified to its high level of floristic originality, the pronounced difference between algal communities and uniqueness. The group of lakes (Nesamovyte, Bolotne Oko, Tsyclop) is characterized by the distinct species composition (in particular the leading taxonomic groups—green, charophyte algae and diatoms). In turn, they are close because of the presence of a significant number of regionally rare species. That could also serve as an additional argument for the protection and preservation of the ecosystem of this waterbody [61,63,65].

*4.3. Ecological Characteristics of the Nesamovyte Lake*

Our ecological investigations published in previous works [61,65] were based on the analysis of selected groups of algae, and the complex species composition is characterized for the first time.

Indication of habitat preferences (with reference to the type of substrate) and changes in the amount of P-B group gives us an assumption that some hydrological changes in the ecosystem of the Nesamovyte Lake has occurred between 1910–1920 and 1967–1978. In turn, the same changes also appeared in the modern period. Such conclusion can be connected also with the focus of investigation into some groups of algae and thus it should be checked by way of further analysis.

Streaming and oxygenation analysis, in the meaning of oxygen concentration and preference of species to mixing of water in lake, also could confirm the changes in oxygen concentration are due to hydrological or hydromorphological changes [112]. However, the small amount of indicator taxa suggests caution before coming to any conclusions. Even though the amount of pseudoaerial forms seems quite strange for the result of the aquatic

investigation and may be connected with the sampling method. The water was sampled in a littoral zone, which is characterized by wave activity or in the ebb and flow zone where high humidity creates certain conditions for the development of a high number of pseudoaerial forms. Analysis of this group of indicators in 1967–1978 showed absence of the organisms that favour streaming water with a high level of oxygenation. This fact can be connected with a focused study of conjugates, however further analysis of indicator taxa in the modern period (2013–2021) testified the stable presence of organisms that favour standing water with low oxygenation. Thus, it also provides a thesis that some changes with the ecosystem of the lake happened.

The characteristics of pH indicator taxa between different years of investigations revealed some changes in water pH. The prevailing of acidophiles that could characterize the acidic conditions in 1910–1920 indicates a marsh type in the studied waterbody (one of the banks had marsh waters) or input of such type of waters into it [112], because this group of algae is characterized by the ability to survive under pH of 5–6. During the 1967–1978 period, the water pH was on the same level, however, the absence of alkaliphiles reveals some changes that have happened at that time. And in the modern period, both hydrological and ecological conditions of the lake have significantly changed. The evidence of it is the high amount of alkaliphiles, testifying to the change of pH to the low-alkaline side under pH 7–8. However, the low number of acidophil taxa is still explained by the presence of marsh type on one bank in the Nesamovyte Lake confirmed by a swampy area with overgrowth of sedge-sphagnum flotation (personal observation). The conducted analysis indicates that the area with marsh conditions has decreased.

Indicator characteristics of salinity displays chloride concentration in the Nesamovyte Lake which allows us to characterize their concentration in the waterbody [112]. As noticeable for all studied periods, the salinity level in the Nesamovyte lake was stable and did not change significantly.

Trophic state analysis [111] indicated changes from an almost minimal level of trophism to some increase in 1967–1978 and a far worst position in 2013–2021. The emergence of hypereutraphetic (he) and an increase in the number of eutraphetic (e) species characterizes the general trend towards the deterioration of the trophic state of the investigated lake. The possible reason for this could be connected to the increasing tourism near the lake.

Characterization of water saprobity indicators as well as saprobity indices made it possible to characterize organic pollution and correlate it with the classification adopted in Ukraine, highlighting water quality classes for the Nesamovyte Lake [113,114]. It revealed the changes in water quality from class I in 1910–1920 to a degradation of the ecological condition of the lake confirmed by the changes in the proportion between classes and appearance of indicators of class IV in 1967–1978 with the same tendency for the deterioration in the modern period.

Thus, the conducted investigation testified a gradual increase in organic pollution from 1910–1920 to the modern period which was also underlined earlier during ecological analyses of the selected taxonomic groups [61,65].

## 5. Conclusions

The comparison between three periods of investigations of the alpine Nesamovyte lake: 1910–1920, 1967–1978 and 2013–2021 revealed significant structural changes in its taxonomic composition of algae, as well as a complex of leading genera and species.

Ecological analysis of the species composition of algae was based on its ecological characteristics and confirmed specific changes that took place with the ecosystem of the lake during different historical periods. The changes in hydrological characteristics, species diversity of algae, systematic structure, communities of dominant and indicator species, the trophic state and organic pollution over a 100-year period revealed the anthropogenic influence in the ecosystem of the lake. The obtained results confirmed the opinion [42,129–131] on the sensitivity and vulnerability of this type of water bodies and its role as an early response-indicator to a change in climate, environmental conditions and an increased level

of anthropogenic impact, as well as the need for efforts to preserve this ecosystem of the Eastern Carpathians.

**Supplementary Materials:** The following are available online at https://www.mdpi.com/article/10.3390/d13060256/s1, Table S1: Species composition and ecological preferences of algae and cyanobacteria of the Nesamovyte Lake in three studied periods (1910–1920; 1967–1978; 2013–2021).

**Author Contributions:** Conceptualization, P.M.T. and O.P.B.; validation, P.M.T., O.P.B., S.B.; investigation, O.M.K.-Z. and H.H.L.; writing—original draft preparation, P.M.T. and O.P.B.: data curation, P.M.T.; writing—review and editing, P.M.T.; visualization, S.B., O.M.K.-Z.; supervision, P.M.T. All authors have read and agreed to the published version of the manuscript.

**Funding:** This research received no external funding.

**Data Availability Statement:** Species composition and ecological preferences of algae and cyanobacteria of the Nesamovyte Lake in three studied periods are available in Supplementary Table S1.

**Acknowledgments:** The authors acknowledge financial and organizational support given by the administration and the employees of the National Academy of Science of Ukraine and Academy of Science of Poland for financial support of the Polish-Ukrainian project of scientific cooperation. We are grateful to I.I. Chorney, V.V. Budjak and O.I. Khudyi (Yuri Fedkovych Chernivtsi University), as well as T. Mykitchak (Institute of Ecology of the Carpathians of the NAS of Ukraine, Lviv) for organizing and conducting field trips to the Ukrainian Carpathians and for assistance in the work. Our gratitude extends to J. Tunovsky (Institute of Freshwater Fish Research, Poland) for the provided hydroecological indicators. Also, authors are very grateful to Taras Kazantsev and Trevor Williams for the proofreading of the English text.

**Conflicts of Interest:** The authors declare no conflict of interest.

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
