# Peer review of "Diversity of Algae and Cyanobacteria and Bioindication Characteristics of the Alpine Lake Nesamovyte (Eastern Carpathians, Ukraine) from 100 Years Ago to the Present"

_diversity, doi:10.3390/d13060256_

Round 1
Reviewer 1 Report
Petro M. Tsarenko, Olena P. Bilous, Olha M. Kryvosheia-Zakharova, Halyna H. Lilitska and Sophia Barinova. Diversity of algae and cyanobacteria and bioindication characteristics of the Alpine Lake Nesamovyte (Eastern Carpathians, Ukraine) from 100 years ago to the present – MS submitted to Diversity
The MS analyzes three periods of the last 100 years based on the study of the algal flora of an Alpine Lake Nesamovyte (Eastern Carpathians, Ukraine). Investigating the vulnerable ecosystems of high mountain lakes is of paramount importance today.
In the chapter material and method for all samplings, from where and how algae were collected should be described in more detail (e.g. „The material for this recent study (18 samples of plankton, periphyton, and squeezes from 115 the moss) was collected along the perimeter of the lake in August 2013–2018. for „The following ecological characteristics were used: Habitat preference, Streaming and oxygenation, pH, Salinity, Trophical state and Сlass of organic pollution.” This is important because these similarities and differences need to be taken into account when evaluating the results. (historical data (1910–1920: WoÅ‚oszyÅ„ska and 1967–1978 collections - collectors Prof. Z.I. Asaul and Prof. G.M. Palamar-Mordvintseva)
The results are partly too textual for ecological characteristics, Figure 7 show it all, so the section between lines 351-456 should be significantly shortened.
In sub-chapter 3.3 Ecological characteristics of the Nesamovyte Lake due to algal preferences –
from line 504 – „The probable reason for the detected diversity of diatoms is also a change in the hydrochemical parameters of the lake – more than 8-fold increase in water salinity over the past half-century (12–99 mg L-1) and a 4-fold increase in the number of sulfates (up to 15.0 mg L-1) and nitrates (up to 2.0 mg L-1) and the resistance ….” In Table 1 and 2 are different data (see below):
Table 1 - Conductivity, μS cm-1, 5.1–6.0,
In a water with so low conductivity no real meaning to discuss about „hl – halophiles, mh – mesohalobes, algae”
This could also be left out of the discussion (lines 593-598): „Indicator characteristics of salinity displays chloride concentration in the Nesamovyte Lake and allows to characterize their concentration in the waterbody [112]. As noticeably for all studied periods the prevailing amount of salinity indicators were presented by indifferents that are typical characteristics of freshwaters. Despite the disappearance of a small group of halophiles in 1967–1978, the salinity level in the Nesamovyte lake was stable and did not change significantly over the studied periods.”

Author Response
- In the chapter material and method for all samplings, from where and how algae were collected should be described in more detail (e.g. „The material for this recent study (18 samples of plankton, periphyton, and squeezes from 115 the moss) was collected along the perimeter of the lake in August 2013–2018. for „The following ecological characteristics were used: Habitat preference, Streaming and oxygenation, pH, Salinity, Trophical state and Сlass of organic pollution.” This is important because these similarities and differences need to be taken into account when evaluating the results. (historical data (1910–1920: WoÅ‚oszyÅ„ska and 1967–1978 collections - collectors Prof. Z.I. Asaul and Prof. G.M. Palamar-Mordvintseva)
Response: Additional information to the material and methods part provided. Some clarification of methodological approaches and detailing of groups of vascular plants and mosses (Carex rostrata Stokes and Sphagnum spp.) in the selection of periphyton and extracts from them have been added. The identity of collection methods for all periods of studying the species diversity of algae of Lake Nesamovite (1910, 1967, 2013) was noted. However, slight differences in presented material with the nuances of focused studies of some groups highlighted each time in the discussion part.
- The results are partly too textual for ecological characteristics, Figure 7 show it all, so the section between lines 351-456 should be significantly shortened.
Response: The results shortened twice times.
- In sub-chapter 3.3 Ecological characteristics of the Nesamovyte Lake due to algal preferences –
from line 504 – „The probable reason for the detected diversity of diatoms is also a change in the hydrochemical parameters of the lake – more than 8-fold increase in water salinity over the past half-century (12–99 mg L-1) and a 4-fold increase in the number of sulfates (up to 15.0 mg L-1) and nitrates (up to 2.0 mg L-1) and the resistance ….” In Table 1 and 2 are different data (see below):
Table 1 - Conductivity, μS cm-1, 5.1–6.0,
Response: Table 1 presents original measurements, while table 2 presents mostly literature data. Perhaps, you were surprised to see the wide range of salinity in earlier times with so usual values of conductivity. This is an interesting note, that was added to the discussion part
- Table 1 - Conductivity, μS cm-1, 5.1–6.0,
In a water with so low conductivity no real meaning to discuss about „hl – halophiles, mh – mesohalobes, algae”
This could also be left out of the discussion (lines 593-598): „Indicator characteristics of salinity displays chloride concentration in the Nesamovyte Lake and allows to characterize their concentration in the waterbody [112]. As noticeably for all studied periods the prevailing amount of salinity indicators were presented by indifferents that are typical characteristics of freshwaters. Despite the disappearance of a small group of halophiles in 1967–1978, the salinity level in the Nesamovyte lake was stable and did not change significantly over the studied periods.”
Response: the discussion of salinity was reduced to the minimum.

Reviewer 2 Report
This is a potentially interesting manuscript that aims at using data collected on three occasions in the last hundred years to illustrate the algal and cyanobacterial diversity of a Ukrainian high-mountain tarn. Unfortunately, as it stands, the manuscript has many shortcomings that need to be fixed before the paper can be considered for publication in Diversity. They are listed below in order of importance:
- The datasets used have several limitations that preclude the possibility of direct comparison among them: the data collected in 1910-1920 suffer the technical limitations of microscopy in that period and, even more, the much more limited taxonomic knowledge and consequent availability of identification keys at that time; it seems that the 1967-1978 data are limited to two algal groups only, which are not the most relevant in this type of environment; recent data are apparently OK but cannot be directly compared to the previous ones for the main reasons outlined above. I would therefore resign to the ambitious goal to document environmental change with the algae and would choose a more realistic goal, such as documenting the algal and cyanobacterial diversity of the tarn using three datasets collected in the last hundred years, each one with its own merits and limitations.
- The paper is too long, verbose, with a lot of speculation based on weak data. I would reduce the text (in particular parts of the Results and the Discussion) to what is really needed for the new goal stated above.
- The Methods are not precise and detailed. The characteristics of the three datasets should be explained in detail. For instance, how and where (in the tarn) were the samples collected on the three occasions? Identification literature used in the three periods.
- The language is very poor. Many sentences are unclear / not understandable. After in-depth revision and shortening the text needs to be edited by a professional language editor or native speaker.
- The terminology is imprecise and/or outdated in several cases, e.g.: * “streaming and oxygenation indicators”: I guess what is meant is indicators of current velocity conditions (still to running waters) and dissolved oxygen concentration [by the way: it is conceptually wrong to use photoautotrophic organisms as indicators of dissolved oxygen concentration]; * “aerophytic” (rather use “pseudo- and euaerial”); * “coastal” is inappropriate for such a small lake (use “shore” or “littoral”); “salinity” not a good term for low-conductivity waters (use “TDS” “Total dissolved Solids”); * “trophicity” (use “trophism” or “trophic state”); * “algosociological”, “algofloristic sub-provinces” etc.: this obsolete terminology based on higher-plant phytosociology and biogeography is inappropriate for algae and cyanobacteria.
- Figure 2: You have data collected on three discrete occasions in the last 100 y not data recorded more or less continuously over the last 100 y, thus please do not use continuous lines but histograms in three groups.
- Figure 3: This analysis doesn’t make sense because the data in the three periods are not comparable (see above).
- Table 1: * With a maximum depth of 2 m this can hardly be defined a lake; it’s more likely to be a polymictic tarn (see e.g. Plant Ecology and Evolution 149: 144–156. http://dx.doi.org/10.5091/plecevo.2016.1206); * the reported water temperatures are very high because only data measured during summer are available (this is, at best, a mean Summer Water Temperature); * These conductivity values are incredibly low (this is commercial distilled water): from what I can estimate from the ionic concentrations in Table 2, your tarn has at least 50-60 µS cm-1 conductivity (please check your data carefully).
- Table 2: Cd, Cr, Pb are typically expressed in µg L-1 not in mg L-1 (please check your data carefully).
- Introduction: Don’t tell that high-mountain lakes are an important topic in limnological congresses but rather cite some recent papers outlining the reasons for their relevance, e.g.: Water 12: 260 DOI: 10.3390/w12010260 (part on high-mountain lakes, pagg. 27-29); Ecological Indicators 125, 107603. DOI: 10.1016/j.ecolind.2021.107603
- The algae defined as “rare” in Figures 4 & 5 are common species in high-mountain lakes and mires.
Author Response
- The datasets used have several limitations that preclude the possibility of direct comparison among them: the data collected in 1910-1920 suffer the technical limitations of microscopy in that period and, even more, the much more limited taxonomic knowledge and consequent availability of identification keys at that time; it seems that the 1967-1978 data are limited to two algal groups only, which are not the most relevant in this type of environment; recent data are apparently OK but cannot be directly compared to the previous ones for the main reasons outlined above. I would therefore resign to the ambitious goal to document environmental change with the algae and would choose a more realistic goal, such as documenting the algal and cyanobacterial diversity of the tarn using three datasets collected in the last hundred years, each one with its own merits and limitations.
Response: Thank you for your idea to set a new aim, it was changed following your advice - This study aims to document the algal and cyanobacterial composition of the Nesamovyte Lake using three datasets collected in the last hundred years and outlining the changes in the ecological state of the lake due to the bioindication characteristics of discovered species composition.
We agree with the reviewer in terms of a certain difficulty in a direct comparison of available data for three periods of research on the diversity of algae in the lake. These difficulties and incompatibility of some data are underlined in the discussion part. At the same time, our work is based on the nature of the change in the species diversity of algae along with the changes in the chemistry of water and some ecological parameters.
The composition of species surviving in a given environment characterizes it based on their ecological preferences. Therefore, we can give a general characterization of the environment based on the ecology of the surviving species.
Some additional explanation to the species composition in three decades and differences in methods: We assume that the methodological approaches for collecting and fixing algae from different ecotopes of the lake were identical in the early and middle of the 20th century, namely, plankton netting, moss pomace (Woloszynska, 1920; Asaul, 1967; Palamar-Mordvintseva, 1978), which allows for comparable comparisons of species composition. The identification of species of most taxonomic groups (except for diatoms) was carried out using a light microscope (LM), which is also comparable to the methods of microscopic study of algae in the 20th century. Using the method of scanning microscopy (SEM) to study the diversity of diatoms contributed to the identification of small-cell forms and clarification of the species identification of some of them. At the same time, in comparative terms, in aggregate, both LM and SEM only confirm the high significance of the Bacillariophyta group in the ecosystem of Lake Nesamovite, and not its absence or secondary importance (e.g., Woloszynska, 1920). This situation is explained by the less used research methods at the beginning of the 20th century. (they are similar or identical, except SEM), and, most likely, the focus of the researcher on other groups of algae. The identification of freshwater algal species of different taxonomic groups, except diatoms, was based on LM-diagnostic characters both in the 20th century and at present (as opposed to diatoms). Thus, the similarity or identity of the methodological approaches to the collection of material, its morphological study and identification make it possible to carry out a correct comparison of the species composition of algae for the studied lake in the historical aspect.
- The paper is too long, verbose, with a lot of speculation based on weak data. I would reduce the text (in particular parts of the Results and the Discussion) to what is really needed for the new goal stated above.
Response: the ecological part in the results is significantly shortened, in the discussion the salinity information was shortened.
- The Methods are not precise and detailed. The characteristics of the three datasets should be explained in detail. For instance, how and where (in the tarn) were the samples collected on the three occasions? Identification literature used in the three periods.
Response: Additional information was added into the materials and methods part.
Also, almost identical methodological approaches for collecting and fixing algae from different ecotopes of the lake were used during the earliest investigation (at the beginning and middle of the 20th century), namely, a plankton net for collecting algae from the water column, squeeze from mosses for distinguishing associated groups of microalgae (Woloszynska, 1920; Asaul, 1967; Palamar-Mordvintseva, 1978). The uniformity of methodological approaches and collection sites (littoral zone of the lake) - ibid) makes it possible for correct comparison of the results of species diversity and obtaining comparative data.
- The language is very poor. Many sentences are unclear / not understandable. After in-depth revision and shortening the text needs to be edited by a professional language editor or native speaker.
Response: the text was given again to the native speaker for additional corrections, improvements may be visible in the text
- The terminology is imprecise and/or outdated in several cases, e.g.: * “streaming and oxygenation indicators”: I guess what is meant is indicators of current velocity conditions (still to running waters) and dissolved oxygen concentration [by the way: it is conceptually wrong to use photoautotrophic organisms as indicators of dissolved oxygen concentration]; * “aerophytic” (rather use “pseudo- and euaerial”); * “coastal” is inappropriate for such a small lake (use “shore” or “littoral”); “salinity” not a good term for low-conductivity waters (use “TDS” “Total dissolved Solids”); * “trophicity” (use “trophism” or “trophic state”); * “algosociological”, “algofloristic sub-provinces” etc.: this obsolete terminology based on higher-plant phytosociology and biogeography is inappropriate for algae and cyanobacteria.
Response: We accept the terminological remarks and are grateful to the referee. The necessary changes and corrections have been made to the text. In particular, the terms “aerophytic” - “pseudoaerial”, “coastal” - “litoral” have been changed. Along with this, one can partially agree with the reviewer regarding the use of the term Salinity, since we use the classification of Van Dam, 1994 and provide a link to his work. Also, the obsolescence of the term "algosozological" remains controversial, however, we accept the comment of the reviewer and replace it with "the questions of protection and preservation of algae". However, the remark about the term "algofloristic subprovinces" is hardly reasoned. The geographic features of the distribution and distribution of algae have not been sufficiently developed, and it is true, as well as the fact that, at the base, their analogy is the terminology that was developed for the geography of higher plants. However, both the term itself and the approaches are used in the development of the algae-zoning of the world ocean and the peculiarities of the distribution of algae in water bodies of land (Krieger, 1932; Zinova, 1962; Vetrova, 1986; Kristiansen, 1996; Coesel, Krienitz, 2008; Klikovskiy, Kuznetsova, 2014; Palamar-Mordvintseva, Tsarenko, 2015 et al.). Proceeding from this, the development of principles and criteria for the geographic zoning of algae, in particular in Europe, is relevant and requires attention, and not transfer to the rank of outdated directions.
- Figure 2: You have data collected on three discrete occasions in the last 100 y not data recorded more or less continuously over the last 100 y, thus please do not use continuous lines but histograms in three groups.
Response: The figure changed according to your recommendations
- Figure 3: This analysis doesn’t make sense because the data in the three periods are not comparable (see above).
Response: Since the methodological and practical approaches to the collection, fixation, processing and identification of the species composition were identical or rather similar, the comparison carried out is indicative, comparable and illustrative. The cladogram graph clearly illustrates the similarity between research and results from the early 20th century. and XXI (clade 1910-1920 and 2013-2021), which were based on the coverage of all groups of algae present in the lake ecosystem. The limitedness of the results of diversity in taxonomic scope is illustrated by the 1967-1978 branch as an isolated clade and having a real explanation (reviewer's text above).
In addition, in the work, considerable emphasis is placed on the ecological characteristics of algae and the analysis of the habitat of the algal flora based on the widely used method of bioindication.
- Table 1: * With a maximum depth of 2 m this can hardly be defined a lake; it’s more likely to be a polymictic tarn (see e.g. Plant Ecology and Evolution149: 144–156. http://dx.doi.org/10.5091/plecevo.2016.1206); * the reported water temperatures are very high because only data measured during summer are available (this is, at best, a mean Summer Water Temperature); * These conductivity values are incredibly low (this is commercial distilled water): from what I can estimate from the ionic concentrations in Table 2, your tarn has at least 50-60 µS cm-1 conductivity (please check your data carefully).
Response: We followed your advice and the possibility to use the term ‘tarn’ for Nesamovyte was added into the text. However, during all period of investigations, there were no researchers who called the studied lake a tarn, thus we decided to leave this term in the text and the title. Moreover, consideration of the “lake” category as a type of water body corresponds to the descriptors of the EU Water Framework Directive (2000) and, in a specific case, is defined as one of the gradations - very small, shallow lakes (with a water surface area <0.5 km2 and roughness <3 m).
Also, we are very grateful for your attentiveness regarding Conductivity measurements. It is a mistake and as it was original measurements, we should delete these values.
- Table 2: Cd, Cr, Pb are typically expressed in µg L-1not in mg L-1 (please check your data carefully).
- Response: Thank you very much, the measurement units
- Introduction: Don’t tell that high-mountain lakes are an important topic in limnological congresses but rather cite some recent papers outlining the reasons for their relevance, e.g.: Water12: 260 DOI: 10.3390/w12010260 (part on high-mountain lakes, pagg. 27-29); Ecological Indicators 125, 107603. DOI: 10.1016/j.ecolind.2021.107603
- Response: The information about congress deleted, and the advised papers added instead of some other sources.
- The algae defined as “rare” in Figures 4 & 5 are common species in high-mountain lakes and mires.
Response: The Figure 4 and 5 were combined and instead one figure with regionally rare diatom species for Ukraine, that were found in the lake were noted. It is important because it is only 4 or fewer records for this species are known for our country. In turn, the noted species presents the arctic flora, which also was highlighted in the text. However, some common species were deleted from the text and they also absent in the new figure.

Reviewer 3 Report
Review of paper entitled “Diversity of algae and cyanobacteria and bioindication characteristics of the Alpine Lake Nesamovyte (Eastern Carpathians, 3 Ukraine) from 100 years ago to the present”
Manuscript presents interesting data about mountain ecosystems of Carpathians. The topic of paper fit to scope of journal and can be publish after making same corrections.
My the most important objection relating to the article is, if I understood correctly - because in methods is not clearly pointed – the lack of re-analysis of historical samples using modern taxonomy. If this really has not been done the statistical analysis has not make sense ! If the authors recognize that they are unable to perform revision of historical data, they should reorganize their manuscript. They should focus on current data and include full list of identified taxa (i.e. as supplementary data). In present form manuscript is definitely too long.
Same additional remarks are noted below:
Introduction:
- L56 - incorrect entry of the unit
Materials and Methods:
- L88 – 1.748 m ? or 1 748 m ?
- Interesting point of study area is protective status of this lake. This area is protected by any low ? National Park ect.?
- Figure 1. – in my opinion you should include subfigures designation (A, B, C ect.) and include it in figure caption. What is the number 2061 ?
I suggest to separate a subsection – study area.
- L92 – 0.3 ha but 88x45 is 3960, what is rather ca. 0.4 ha.
- Table 1 table 2 – I suggest move these tables to results or include their content to text and delete tables.
- L103,104 – italics for Latin names
- Line 115, 116 – 18 samples per year or per 2013-2018? What was the total number of samples in 2013-2018? What was the total number of samples analyzed during whale studies.
- L124 – use subscript and superscript for chemical formula.
- L124, 125 – “and an Olympus BX-53 (lens ×100 immersion)” - redundant here
- L127 – “the samples were applied to special brass tables” – specimen stubs ?
- L127,128 - “sprayed 127 with gold” - specify layer thickness and coater type.
- JSM-6060LA – use full name of microscope
- L137, 138 – What collection? Very important aspect of this studies! Authors write about 100 years comparison and provide no detailed information about it. Where come the historical data (samples) from ? Based on what the authors made taxonomical comparison? Did authors verify the archive microscope slides (samples) or did they only compare their results with literature data? This is one of the most important aspect of the work ! In Line 153–156 Authors cite the include references for whale study period (1910 – 2021). The question is like above – did Authors made the taxonomical companion base on the original samples ? If NOT, they should do this for comparison base on recent literature.
Results
- Table 3 – What mean period 1920-2021 – from were came those data?
- Line 177–180 genera names in italics.
- Figure 2 – the results base on “original data” or are results of taxonomical verification ?
- Figure 3 – The one important question to results and methods – how the statistics were made ? On Figure 3 I see three period (samples) – does mean the analysis were made base on only 3 samples ? Authors should made this companioning base on ALL studied samples, not on averaged values for three periods.
- L255-260, Figure 4 – I don’t see any reason to highlight those species in the paragraph and on Figure 4. These taxa are quite common worldwide.
- L318-321 delete space between number and % symbol.
- L351 Habitat or habitat.
- The ecological preferences analysis is definitely too long !– Authors should focus considerations on the most important aspects, i.e. main differences, similarities, or species with the most important influence on ecological characterization of lake.
Discussion:
- L501, 502 – yes, but base on this sentence I conclude that Authors did not made taxonomical revision of historical samples. If I have right Authors should do this because since 1910’s the taxonomy of algae has change drastically.
- L507-509 – use superscript or multiplication symbol between number and unit.
- Discussion is too long, too much repetitions of results.
Conclusions are too long and too general.
Author Response
My the most important objection relating to the article is, if I understood correctly - because in methods is not clearly pointed – the lack of re-analysis of historical samples using modern taxonomy. If this really has not been done the statistical analysis has not make sense ! If the authors recognize that they are unable to perform revision of historical data, they should reorganize their manuscript. They should focus on current data and include full list of identified taxa (i.e. as supplementary data). In present form manuscript is definitely too long.
Response: Almost identical methodological approaches for collecting and fixing algae from different ecotopes of the lake were used during the earliest investigation (at the beginning and middle of the 20th century), namely, a plankton net for collecting algae from the water column, squeeze from mosses for distinguishing associated groups of microalgae (Woloszynska, 1920; Asaul, 1967; Palamar-Mordvintseva, 1978). The uniformity of methodological approaches and collection sites (littoral zone of the lake) - ibid) makes it possible for correct comparison of the results of species diversity and obtaining comparative data. The revision of historical data, as well as re-analysis, have been made and the comparison with current data have been provided. Some parts of the manuscript shortened.
Same additional remarks are noted below:
Introduction:
- L56 - incorrect entry of the unit
Response: Changed
Materials and Methods:
- L88 – 1.748 m? or 1 748 m?
Response: Thanks, 1 748 m, changed.
- Interesting point of study area is protective status of this lake. This area is protected by any low? National Park ect.?
Response: The lake is located in Carpathian National Nature Park. This information was added into the text.
- Figure 1. – in my opinion you should include subfigures designation (A, B, C ect.) and include it in figure caption. What is the number 2061? I suggest to separate a subsection – study area.
Response: The subsections have been added to the picture and explained in legend for the figure. 2 061 m is the highest point of Ukraine provided for the Hoverla mount.
L92 – 0.3 ha but 88x45 is 3960, what is rather ca. 0.4 ha.
Response: Thanks, changed.
- Table 1 table 2 – I suggest move these tables to results or include their content to text and delete tables.
Response: The tables are combined, and it is better to leave the combined table in the Material and methods part, where we mention the data from the table for the first time.
L103,104 – italics for Latin names
Response: done
- Line 115, 116 – 18 samples per year or per 2013-2018? What was the total number of samples in 2013-2018? What was the total number of samples analyzed during whale studies.
Response: per 2013-2018, changed. The number of samples for the 1967–1978 period equal 44 and this information was added into the text. Unfortunately, it is impossible to know the number of samples from 1910–1920 years as this information was not provided by the author. Altogether 62 samples + ? were analysed for this paper.
- L124 – use subscript and superscript for chemical formula.
Response: done
- L124, 125 – “and an Olympus BX-53 (lens ×100 immersion)” - redundant here
Response: If we understood you right, the information regarding immersion and lens was deleted.
- L127 – “the samples were applied to special brass tables” – specimen stubs?
Response: specimen stubs. Thank you
- L127,128 - “sprayed 127 with gold” - specify layer thickness and coater type. JSM-6060LA – use full name of microscope
Response: Additional information provided to the manuscript: (10-20 nm) JFC-1600 sputter coater, and examined in a scanning electron microscope JSM-6060LA (JEOL, Japan)
- L137, 138 – What collection? Very important aspect of this studies! Authors write about 100 years comparison and provide no detailed information about it. Where come the historical data (samples) from ? Based on what the authors made taxonomical comparison? Did authors verify the archive microscope slides (samples) or did they only compare their results with literature data? This is one of the most important aspect of the work ! In Line 153–156 Authors cite the include references for whale study period (1910 – 2021). The question is like above – did Authors made the taxonomical companion base on the original samples ? If NOT, they should do this for comparison base on recent literature.
Response: The material for this study is based on the analysis of first record (1910–1920) for the Nesamovyte Lake published by WoÅ‚oszyÅ„ska, work with samples from 1967–1978 (collectors Prof. Z.I. Asaul and Prof. G.M. Palamar-Mordvintseva) and our own investigations (2013–2018).
The species list provided by WoÅ‚oszyÅ„ska (collected in 1910) [67] was carefully checked and analysed to describe the period of 1910–1920.
For period of 1967–1978 the samples from Algoteca funds of M.G. Kholodny Institute of Botany of NAS of Ukraine (AKW) – NN 16855–16898 (samples from the year 1967–1978, collectors Prof. Z.I. Asaul and Prof. G.M. Palamar-Mordvintseva) were studied (44 samples). In turn, the published data [2–5] from the period of 1967–1978 based on these samples also was included into current analysis.
To characterize modern period (2013–2021), 18 samples of net plankton (50-100 of water filtered out), periphyton from vascular plants (dead and living parts of herbaceous plants (Carex rostrata) and pine branches (Pinus mugo Turra) and squeezes from the moss (Sphagnum spp.) were collected along the perimeter of the lake in August 2013–2018. This study is based on the living algal material from plankton (algal cultures, with the addition of BBM-medium) [78] and samples that were fixed with 4% formaldehyde solution.
The obtained results seem to be comparable as almost identical methods for collecting and fixation of the algal material for all periods of studying the species diversity of algae of Lake Nesamovite (1910, 1967, 2013) were used.
We clarified this in the material and method section of the manuscript. The species lists of all periods were validated using Algaebase system (also added into the M&M section).
- Table 3 – What mean period 1920-2021 – from were came those data?
- Response: Thanks, our mistake. Period of 1910–2021, that occupies all data that were found for the Nesamovyte during all periods of studies.
- Line 177–180 genera names in italics.
- Response:
- Figure 2 – the results base on “original data” or are results of taxonomical verification ?
Response: The Figure 2 is the mix of our data and literature data that were verified. We’ve added the explanation in the M&M section as well as the link in the Results section
The taxonomic structure of the algal composition of the Nesamovyte Lake is a summary of the available data on the species composition of algae for 100 years of investigations during the following study periods: 1910–1920, 1967–1978 to 2013–2021 [2, 4, 61, 63, 65, 67, 106, 121]. It is represented by 234 species (245 infraspecific taxa (inft) composing 8 divisions, 15 classes, 33 orders, 55 families, 100 genera of cyanobacteria and algae (Table 2, Figure 2).
Figure 3 – The one important question to results and methods – how the statistics were made ? On Figure 3 I see three period (samples) – does mean the analysis were made base on only 3 samples ? Authors should made this companioning base on ALL studied samples, not on averaged values for three periods.
Response: The necessary information was added into the M&M section:
The Cluster analysis of algal composition was carried out using the Paleontological Statistics Software (PAST) to measure the degree to which species composition was similar among the studied periods (1910–1920, 1967–1978, 2013–2021) [119]. For this purpose, the presence/absence data were used in the meaning of the Sørencen coefficient, calculated in the program as Bray-Curtis Similarity index [120].
- The statistics were made according to the samples for each period (WoÅ‚oszyÅ„ska didn’t provide the exact number, however, the data list is based on some amount of samples not one), 44 materials from Algothece from the Institute of botany studied by us along with the species list from published sources (papers 3- 5 in the References), and original data (18 samples).
- L255-260, Figure 4 – I don’t see any reason to highlight those species in the paragraph and on Figure 4. These taxa are quite common worldwide.
Response: The Figure 4 and 5 were combined and instead one figure with regionally rare diatom species for Ukraine, that were found in the lake were noted. It is important because it is only 4 or fewer records for this species are known for our country. In turn, the noted species presents the arctic flora, which also was highlighted in the text. However, some common species were deleted from the text and they also absent in the new figure.
New part of the text:
‘Moreover, for the modern period of studies in the Nesamovyte Lake, the presence of arctic diatom species that also are regionally rare for Ukraine [105], in particular, Cavinula pseudoscutiformis (Hustedt) D.G. Mann & Stickle, Pinnularia rhombarea Krammer, P. rupestris Hantzsch, P. subanglica Krammer were noted (Figure 4).’
- L318-321 delete space between number and % symbol.
- Response: Thanks, done for these lines and the same cases in the text.
- L351 Habitat or habitat.
- Response: The names of categories were decided to write form capital letter
- The ecological preferences analysis is definitely too long !– Authors should focus considerations on the most important aspects, i.e. main differences, similarities, or species with the most important influence on ecological characterization of lake.
- Response: The ecological analysis was shortened following the advice.
Discussion:
- L501, 502 – yes, but base on this sentence I conclude that Authors did not made taxonomical revision of historical samples. If I have right Authors should do this because since 1910’s the taxonomy of algae has change drastically.
- Response: The newest updated lists were used for the comparison. To clarify this we add the explanation to the materials and methods part.
- Identified taxa, as well as all algal species lists from previous years of studies of the territory were validated using the AlgaeBase system [104] and “Algae of Ukraine …” [105] monographic series.
- L507-509 – use superscript or multiplication symbol between number and unit.
Response: done
- Discussion is too long, too much repetitions of results.
Response: The repetitions from the results were deleted. Some parts of the Discussion were rewritten
Conclusions are too long and too general.
Response: The conclusions were rewritten and shortened

Reviewer 4 Report
Tsarenko, P.M. et al. Diversity of algae and cyanobacteria and bioindication characteristics of the Alpine Lake Nesamovyte (Eastern Carpathians, Ukraine) from 100 years ago to the present – submitted paper
The study of the high mountain lakes of the Carpathians is very important in aspect of conservation of biodiversity. This is a unic paper on this topic analyzing the last 100-year algological study of an alpine lake.
It is a difficult task to compare the results published 100 years ago by reanalyzing samples collected in the 1960s and 70s and the new LM and SEM analyzis of recent samples. A dozen excellent LM and SEM micrographs shows the rare diatom species of Lake Nesamovyte, to help the better understanding the interesting results.
It is difficult to understand the Figure 3. the Bray-Curtis tree of species composition comparison in three periods of the Nesamovyte 198 Lake study (1910–1920; 1967–1978; 2013–2021). A more thorough explanation would be needed here.
The description of „Ecological characteristics of the Nesamovyte Lake due to algal preferences” is too long, it needs to be shortened and compressed.
Author Response
biodiversity. This is a unic paper on this topic analyzing the last 100-year algological study of an alpine lake.
It is a difficult task to compare the results published 100 years ago by reanalyzing samples collected in the 1960s and 70s and the new LM and SEM analyzis of recent samples. A dozen excellent LM and SEM micrographs shows the rare diatom species of Lake Nesamovyte, to help the better understanding the interesting results.
Response: The material for this study is based on the analysis of first record (1910–1920) for the Nesamovyte Lake published by WoÅ‚oszyÅ„ska, work with samples from 1967–1978 (collectors Prof. Z.I. Asaul and Prof. G.M. Palamar-Mordvintseva) and our own investigations (2013–2018).
The species list provided by WoÅ‚oszyÅ„ska (collected in 1910) [67] was carefully checked and analysed to describe the period of 1910–1920.
For period of 1967–1978 the samples from Algoteca funds of M.G. Kholodny Institute of Botany of NAS of Ukraine (AKW) – NN 16855–16898 (samples from the year 1967–1978, collectors Prof. Z.I. Asaul and Prof. G.M. Palamar-Mordvintseva) were studied (44 samples). In turn, the published data [2–5] from the period of 1967–1978 based on these samples also was included into current analysis.
To characterize modern period (2013–2021), 18 samples of net plankton (50-100 of water filtered out), periphyton from vascular plants (dead and living parts of herbaceous plants (Carex rostrata) and pine branches (Pinus mugo Turra) and squeezes from the moss (Sphagnum spp.) were collected along the perimeter of the lake in August 2013–2018. This study is based on the living algal material from plankton (algal cultures, with the addition of BBM-medium) [78] and samples that were fixed with 4% formaldehyde solution.
The obtained results seem to be comparable as almost identical methods for collecting and fixation of the algal material for all periods of studying the species diversity of algae of Lake Nesamovite (1910, 1967, 2013) were used.
We clarified this in the material and method section of the manuscript. The species lists of all periods were validated using Algaebase system (also added into the M&M section).
It is difficult to understand the Figure 3. the Bray-Curtis tree of species composition comparison in three periods of the Nesamovyte Lake study (1910–1920; 1967–1978; 2013–2021). A more thorough explanation would be needed here.
Response: This part was rewritten and also the details are provided in the material and methods part.
The description of „Ecological characteristics of the Nesamovyte Lake due to algal preferences” is too long, it needs to be shortened and compressed.
Response: This part was shortened and compressed
Round 2
Reviewer 2 Report
The Authors did their best to revise the ms.
Author Response
Dear Reviewer, thank you so much for your valuable comments. They helped us to improve our manuscript.
Reviewer 3 Report
Dear Authors,
The paper “Diversity of algae and cyanobacteria and bioindication charac-2 teristics of the Alpine Lake Nesamovyte (Eastern Carpathians, 3 Ukraine) from 100 years ago to the present” still need same important improvements.
The most important are listed below:
Discussion is still too long and concert many of topics not really connected. I have one advice and request for Authors – they should do additional statistics (the Figure 3 is useless !) for all samples (base on quantitative data, not only zero-one records) from years 1967-2018 base on individual samples (all samples should be visible in graph), and focus paper consideration on this analysis. They can do simple Ward dendrogram or PCA analysis. Base on those results they should compare 50 years of studies. WoÅ‚oszyÅ„ska’s data should use only in introduction and discussion.
L 124-127 – Very important think but still unclear. Do Authors re-analyze original samples using LM microscope or only studied literature data?
L129 “To characterize the modern period (2013–2021)” – but only years 2013-2018 were studied not 2013-2021. Please change to 2013-2018.
L136-137 – is not really true ! If Authors reanalyze original samples from 1967-1978 and samples form 2013-2018, this sentence is refers to only these two periods. The same sampling methods don’t give the reliable result, the identification of algae is even more important.
Authors had no access to original WoÅ‚oszyÅ„ska’s samples so should focus on period since 1967 and use WoÅ‚oszyÅ„ska’s data for general comparisons and discussion.
L142 – preparations = slides
Table 1 – what is: TSS, TDS?
Table 2 – last samples were collected in 2018 so in table should be 1910-1918 ? Add abbreviation to table: Sp, inf.
How you explain no diatom taxa in 1967-1978? This sis very important and interesting issue.
Methods- I still don’t find information about data analysis – statistics. Please add this information. Do Authors use all samples, combine samples form different substrates to one, ect…..? Those information should be included to methods.
L230 – 2013-2021 or 2013-2018?
L 265, 434 – as above
L336-341 base on what kind of data dendrogram has been created? No information in methods.
L 463-471- Here is one problem. In line 124-127 Authors write that samples from years 1967-1978 were analyzed – that mean they made original sample verification so problem from line 467-471 don’t make sense. Additionally Author mention about invasive species - what they mean? Please includes the names of those invasive species. If they analyze samples they should identified those species.
L 444–458 useless here – this information is included in “supplementary”, you can cite it here.
L 473-475 – that TDS is not “pure” salinity. Those statements have aims to explain the low number of diatoms ? If yes, is not true! Diatoms can live in very extreme habitats and this level of ions is not problem.
Author Response
Dear Authors,
The paper “Diversity of algae and cyanobacteria and bioindication charac-2 teristics of the Alpine Lake Nesamovyte (Eastern Carpathians, 3 Ukraine) from 100 years ago to the present” still need same important improvements.
The most important are listed below:
Discussion is still too long and concert many of topics not really connected. I have one advice and request for Authors – they should do additional statistics (the Figure 3 is useless !) for all samples (base on quantitative data, not only zero-one records) from years 1967-2018 base on individual samples (all samples should be visible in graph), and focus paper consideration on this analysis. They can do simple Ward dendrogram or PCA analysis. Base on those results they should compare 50 years of studies. WoÅ‚oszyÅ„ska’s data should use only in introduction and discussion.
Response: We are grateful to the reviewer for the attention and further in-depth discussion of the results of the manuscript. However, additional comments require, apparently, further explanation and attention to the presented material of the article. In the first reply to the reviewer, we explained the necessity, in our opinion, of Figure 3 and its expediency in this article. We disagree with the opinion of the reviewer about excluding WoÅ‚oszyÅ„ska’s data from the discussion since this is the basic information necessary to understand the dynamics of the total diversity of algae in the studied reservoir. It is these data that form a comparative analysis of the species diversity of algae over 100 years. The results of research on the species composition of the early XX century illustrate the state of the algal flora of the reservoir and its taxonomic structure before the period of increased anthropogenic pressure on the reservoir in modern conditions (this is a key point and the rejection of it leads to the loss of the significance of the entire floristic-taxonomic analysis in the article). Along with this, it should be noted that the data on the species composition of algae that are presented in the article WoÅ‚oszyÅ„ska's paper (1920) were critically analysed, nomenclature-taxonomically adapted for all groups of algae and brought to the modern understanding of the status of specific taxa, as well as their belonging to taxonomic groups of high rank, i.e. adapted to the current understanding of taxonomic units.
It is impossible to agree with the focus of the discussion only on the data of 1967-1978. These are data from an incomplete analysis of the species diversity of algae in the reservoir, but only of individual groups - euglena and desmidiaceae (Tsarenko et al., 2019), without data on the composition of diatoms and other groups noted by Wołoszyńska (1920) or under modern conditions of study (data of this manuscript). At the same time, it was the data of this period that made it possible to compare the diversity of these taxonomic groups with the previous period of research in the work of Wołoszyńska (1920) and the 50 years (Asaul, 1967; Palamar-Mordvintseva, 1978, 1982) with addition and an indication of differences in the present (Tsarenko et al., 2019). The important thing is the transformation of the species composition of algae not over 50 years, but in dynamics for the total diversity over 100 years.
L 124-127 – Very important think but still unclear. Do Authors re-analyze original samples using LM microscope or only studied literature data?
Response:
In the section Materials and methods (line 118-138) it is indicated that the authors of the article analyzed the literature data of the early 20th century, as well as the period 1967-1978, with an additional study of Algoteka's samples to the Institute of Botany of the National Academy of Sciences of Ukraine on euglena and desmidiaceae (collectors prof. Z.Asaul and G. Palamar-Mordvintseva) for the last period. For this, similar LM methods and approaches to the identification of algae of the indicated groups - euglena and desmidiaceae algae - were used.
L129 “To characterize the modern period (2013–2021)” – but only years 2013-2018 were studied not 2013-2021. Please change to 2013-2018.
Response: It is impossible to agree with the remark of the reviewer about the chronology of the period of study of the material. This is a misconception understanding and remark. The article reliably indicates the period 2013-2021. study of algological material in the laboratory of the Institute of Botany (see the section Materials and methods) until February 2021 (before the submission of materials to 1 international conference). However, the collection of algological samples was carried out in the period 2013-2018.
L136-137 – is not really true ! If Authors reanalyze original samples from 1967-1978 and samples form 2013-2018, this sentence is refers to only these two periods. The same sampling methods don’t give the reliable result, the identification of algae is even more important.
Authors had no access to original WoÅ‚oszyÅ„ska’s samples so should focus on period since 1967 and use WoÅ‚oszyÅ„ska’s data for general comparisons and discussion.
Response:
It is difficult to agree with the comment of the reviewer (see the response to paragraph 1 of the comments). It is precisely similar approaches in the methodological plan of collecting and analysing species diversity, as well as similar approaches to identifying species using LM microscopy (as at the beginning of the 20th century, as well as in the middle of the 20th century) that allow a correct comparison of the species composition (it is not is about the study of diatoms and an emphasis on the SEM-diagnostics of species).
It is also wrong to refuse when comparing the diversity, from the date of the research results of the beginning of the 20th century. (Woloszynskaja, 1920). These are the initial data for comparison, which are adapted to the modern nomenclature-taxonomic understanding of the volume and structure of the species. It is important to carefully analyse the diversity of algae (with similar study approaches) over 100 years. Fortunately, such data are available and rejection of them is an erroneous methodological approach.
L142 – preparations = slides
Response: changed
Table 1 – what is: TSS, TDS?
Response: added the full names Total Suspended Solids, Total Dissolved Solids
Table 2 – last samples were collected in 2018 so in table should be 1910-1918 ? Add abbreviation to table: Sp, inf.
How you explain no diatom taxa in 1967-1978? This sis very important and interesting issue.
Methods- I still don’t find information about data analysis – statistics. Please add this information. Do Authors use all samples, combine samples form different substrates to one, ect…..? Those information should be included to methods.
Response: Similarly, as indicated by Wołoszyńska's results (1920), which processed the samples collected by prof. Raciborski in 1910, we mark the period of modern sampling (2013-2018) and the end of the period of their identification (2021, see answer above). The abbreviation Sp (inft) means that in that column you may find such figures 15 (16) with the sense that 15 species and 15 infraspecies taxa. If to change it into a coma, it will lose sense.
Unfortunately, data on the diversity of this group of algae are absent in the 1967-1978 literature. (Asaul, 1967; Palamar-Mordvintseva, 1978, 1982), therefore, attention is not focused on the diversity of diatoms and the possible nature of changes in their species composition in comparison with 1920 is not discussed, but they were only mentioned there when discussing the general diversity (Woloszynskaja, 1920).
The procedure for sample handling and identification of the diatom composition is noted in the Materials and Methods section and Article 65. Kryvosheia, O.M .; Tsarenko, P.M. 2018 - Int. J. on Algae 20, pp. 239-264, https://doi.org/10.1615/InterJAlgae.v20.i3.40. Information on the use of a pooled sample to study the diversity of diatoms is included in the Materials and Methods section.
L230 – 2013-2021 or 2013-2018?
L 265, 434 – as above
Response: We explained it already several times.
L336-341 base on what kind of data dendrogram has been created? No information in methods.
Response: The character of the dendrogram construction is mentioned in the Materials and Methods section:
The Cluster analysis of algal composition was carried out using the Paleontological Statistics Software (PAST) to measure the degree to which species composition was similar among the studied periods (1910–1920, 1967–1978, 2013–2021) [119]. For this purpose, the presence/absence of data were used in the meaning of the Sørencen coefficient, calculated in the program as Bray-Curtis Similarity index [120].
L 463-471- Here is one problem. In line 124-127 Authors write that samples from years 1967-1978 were analyzed – that mean they made original sample verification so problem from line 467-471 don’t make sense. Additionally Author mention about invasive species - what they mean? Please includes the names of those invasive species. If they analyze samples they should identified those species.
Response: We consider lines 467-471 as necessary and in them we explain not only the data in 1967-78, but also discuss all studied periods. In turn, we are grateful to our reviewer and we change the invasive species into mesotrophic and eutrophic species
L 444–458 useless here – this information is included in “supplementary”, you can cite it here.
Response: The presented species were added to highlight the common names for 3 mountain systems, especially green coccoids (that are not common in mountains) and some charales. This information is limited in scientific papers and we hope that this information is interesting for readers.
L 473-475 – that TDS is not “pure” salinity. Those statements have aims to explain the low number of diatoms? If yes, is not true! Diatoms can live in very extreme habitats and this level of ions is not problem.
Response: The «salinity» was changed into TDS. We realize that not the TDS level is the reason of changes in number of diatoms, but such impressive change in TDS also reflects total ecological changes in the lake. Thus, we consider this information as necessary.